# Loss of the multifunctional RNA-binding protein RBM47 as a source of selectable metastatic traits in breast cancer

Sakari Vanharanta[1,2†], Christina B Marney[3†], Weiping Shu[1], Manuel Valiente[1], Yilong Zou[1], Aldo Mele[3], Robert B Darnell[3,4*], Joan Massagué[1,5*]

[1]Cancer Biology and Genetics Program, Memorial Sloan-Kettering Cancer Center, New York, United States; [2]MRC Cancer Unit, University of Cambridge, Cambridge, United Kingdom; [3]Laboratory of Molecular Neuro-Oncology, The Rockefeller University, New York, United States; [4]Howard Hughes Medical Institute, The Rockefeller University, New York, United States; [5]Howard Hughes Medical Institute, Memorial Sloan-Kettering Cancer Center, New York, United States

**Abstract** The mechanisms through which cancer cells lock in altered transcriptional programs in support of metastasis remain largely unknown. Through integrative analysis of clinical breast cancer gene expression datasets, cell line models of breast cancer progression, and mutation data from cancer genome resequencing studies, we identified *RNA binding motif protein 47 (RBM47)* as a suppressor of breast cancer progression and metastasis. RBM47 inhibited breast cancer re-initiation and growth in experimental models. Transcriptome-wide HITS-CLIP analysis revealed widespread RBM47 binding to mRNAs, most prominently in introns and 3′UTRs. RBM47 altered splicing and abundance of a subset of its target mRNAs. Some of the mRNAs stabilized by RBM47, as exemplified by *dickkopf WNT signaling pathway inhibitor 1*, inhibit tumor progression downstream of RBM47. Our work identifies RBM47 as an RNA-binding protein that can suppress breast cancer progression and demonstrates how the inactivation of a broadly targeted RNA chaperone enables selection of a pro-metastatic state.

*For correspondence: darnelr@ rockefeller.edu (RBD); j-massague@ski.mskcc.org (JM)

†These authors contributed equally to this work

## Introduction

Cancers arise through an evolutionary process that feeds from stochastic genetic alterations and selection (*Vogelstein et al., 2013*). The identities of the alterations that get selected for are rapidly coming to light through large-scale resequencing efforts. For example, several independent studies have characterized the mutational complement of breast cancer, one of the most common human malignancies (*Shah et al., 2009*; *Stephens et al., 2009*; *Ding et al., 2010*; *Banerji et al., 2012*; *Cancer Genome Atlas Network, 2012*; *Shah et al., 2012*; *Stephens et al., 2012*). Besides confirming previously known cancer genes, such as *TP53* and *PIK3CA*, most studies also report a long tail of rarely mutated genes. While many of these mutations are likely to be passenger events, some of them are potential mediators of tumor phenotypes. How to identify such low-frequency driver mutations remains a challenge.

In addition to mutations that directly promote tumorigenesis through specific alterations in cell signaling and repair pathways, many aberrations found in cancers do not affect cell signaling pathways directly, but rather, they support the stabilization of altered transcriptomic profiles that facilitate the emergence of pro-tumorigenic and metastatic traits. Mutations in epigenetic regulators fall into this category (*Shen and Laird, 2013*). The phenotypic output of such alterations would depend on the activity of already existing signaling processes. Indeed, examples of epigenetic alterations

**eLife digest** Tumors form when mistakes in the genes of a single cell allow it to multiply uncontrollably. Sometimes further mutations in genes allow the cancerous cells to escape from the tumor, enter the bloodstream and start a second cancer elsewhere in the body. However, many of the genetic changes behind this process, which is called metastasis, are poorly understood—especially those changes in genes that occur rarely, but can still help the cancer to spread.

Vanharanta, Marney et al. have looked at data on which genes are switched 'on' or 'off' in metastatic breast cancer cells. A gene called *RBM47* was often switched off in these cells, and patients with a low level of *RBM47* tended to have a poor clinical outcome.

To test the function of the gene, Vanharanta, Marney et al. switched on *RBM47* in cancer cells that had spread from the breast to either the lungs or the brain, and then injected these cells into mice. Few of these cells were able to invade lung and brain tissues in the mice. However, switching off the *RBM47* gene in breast cancer cells had the opposite effect; these cells invaded the lungs of mice more efficiently.

*RBM47* encodes a protein that sticks to molecules of messenger RNA: molecules that transport the instructions encoded in DNA to the machinery that builds proteins. Vanharanta, Marney et al. found that the wild-type RBM47 protein increased the levels of 102 different messenger RNA molecules, but decreased the levels of another 92. Further experiments showed that RBM47 also slows the rate at which messenger RNA molecules are broken down inside cells: this results in the accumulation of proteins that slow down the growth of tumors. Without RBM47, tumor growth is unleashed. Further work is needed to test if increasing RBM47 activity could be used as a new treatment for some types of cancer.

that result in a phenotypic trait in the presence of a specific transcriptional program have been described (*Vanharanta et al., 2013*). Analogously, aberrations in the multi-step mRNA processing and turnover cascade (*Moore and Proudfoot, 2009*) could also lock in aberrant transcriptomic states.

Precise regulation of RNA metabolism is fundamental in the generation of biological complexity in both normal and disease states (*Sharp, 2009*; *Licatalosi and Darnell, 2010*). The concerted action of multiple RNA binding proteins (RBPs) regulate the spatial, temporal and functional dynamics of the transcriptome via alternative splicing, alternative polyadenylation and transcript stability (*Moore and Proudfoot, 2009*). While malignancy-associated dysregulation of RNA metabolism via aberrant microRNA expression is relatively well established (*Di Leva et al., 2013*; *Pencheva and Tavazoie, 2013*), a growing body of evidence indicates a prominent role also for RBPs in both the development and progression of cancer. For example, upregulation of the splicing regulator SRSF1 is associated with multiple tumor types (*Karni et al., 2007*), and is necessary for the oncogenic activity of MYC in lung cancer (*Das et al., 2012*). Conversely, RBM5 has been shown to be tumor suppressive in several cancer models (*Oh et al., 2002*; *Mourtada-Maarabouni et al., 2006*; *Oh et al., 2006*).

We hypothesized that combining cancer genome resequencing data with gene expression information from both clinical data sets and experimental model systems of metastasis would allow the identification of rarely mutated cancer genes with potential functional significance. This approach identified *RNA binding motif protein 47 (RBM47)* as a suppressor of breast cancer progression. By analyzing the transcriptome-wide RBM47 binding patterns we demonstrate that RBM47, a previously uncharacterized RNA-binding protein, modulates mRNA splicing and stability. Loss of RBM47 function thus provides a specific example of the power of global RNA modulatory events in the selection of pro-metastatic phenotypic traits.

## Results

### RBM47 inactivation associated with breast cancer progression

We combined gene expression data from triple negative metastatic breast cancer models (*Minn et al., 2005*; *Bos et al., 2009*) and a cohort of 368 untreated clinical breast cancer cases (*Minn et al., 2005*;

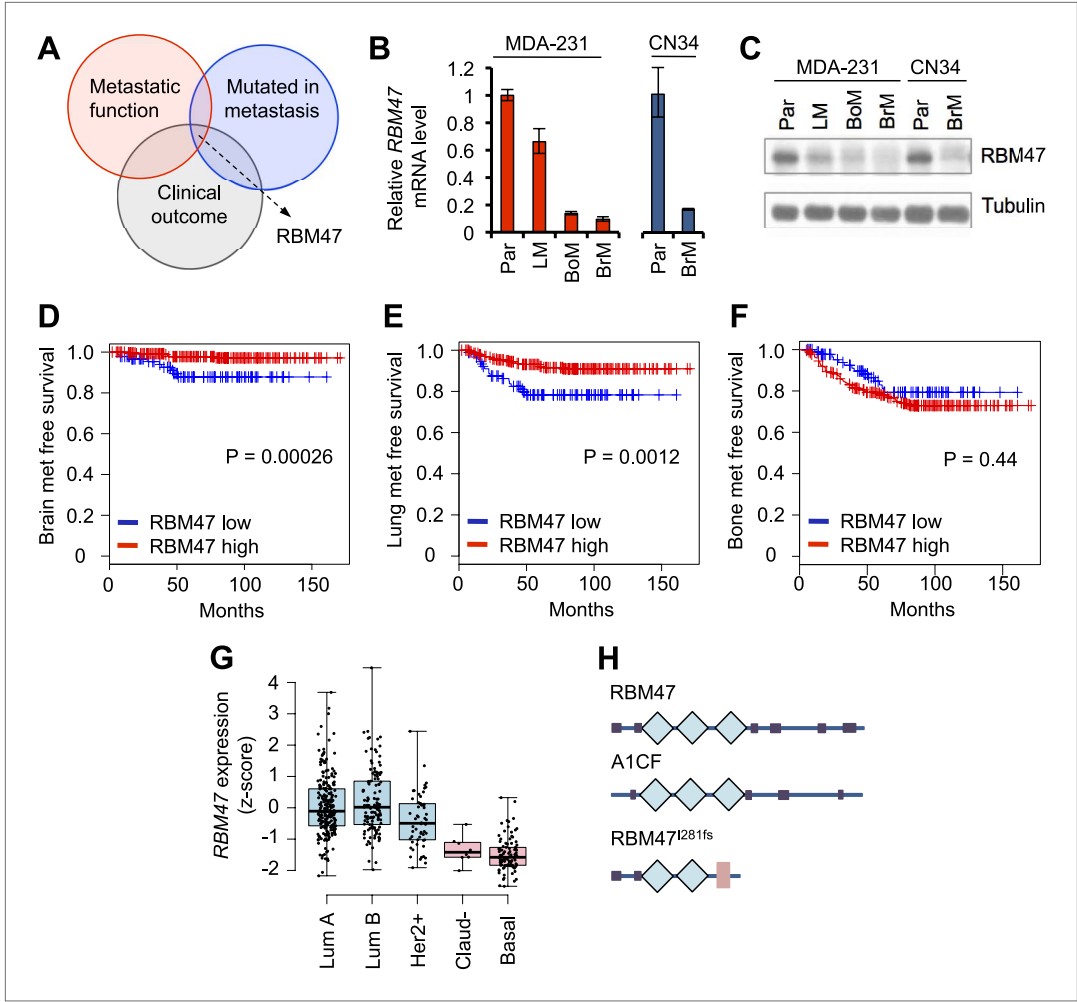

**Figure 1**. RBM47 expression associated with breast cancer progression. (**A**) A schematic of the analytical approach. Genes identified as mutated in a breast cancer brain metastasis by *Ding et al. (2010)* where compared to metastasis-associated gene expression traits in both clinical data sets and experimental model systems. This identified *RBM47* as a putative breast cancer suppressor gene. (**B**) *RBM47* mRNA expression measured by quantitative real-time RT-PCR in two cell line systems of breast cancer metastasis. In both panels, the data are normalized to the parental cell line (Par). Error bars represent 95% confidence intervals obtained from multiple PCR reactions. LM, lung metastatic derivative; BoM, bone metastatic derivative; BrM, brain metastatic derivative. (**C**) RBM47 protein expression measured by Western blotting. Samples as in (**B**). Tubulin used as a loading control. (**D**) Brain metastasis free survival in a cohort of 368 untreated breast cancer patients. Cases classified based on *RBM47* mRNA expression, bottom 1/3 in blue, top 2/3 in red. p-value derived from a Cox proportional hazards model using *RBM47* expression as a continuous variable. (**E**) Lung metastasis free survival in a cohort of 368 untreated breast cancer patients. Cases classified based on *RBM47* mRNA expression, bottom 1/3 in blue, top 2/3 in red. p-value derived from a Cox proportional hazards model using *RBM47* expression as a continuous variable. (**F**) Bone metastasis free survival in a cohort of 368 untreated breast cancer patients. Cases classified based on *RBM47* mRNA expression, bottom 1/3 in blue, top 2/3 in red. p-value derived from a Cox proportional hazards model using *RBM47* expression as a continuous variable.(**G**) RBM47 expression as measured by RNA-seq in the TCGA data set of 748 patients. Samples grouped based on breast cancer molecular subtype: luminal A, luminal B, Her2 positive, claudin low and basal. RBM47 expression is lower in claudin low and basal subtypes, both of which are associated with poor patient outcome. (**H**) A schematic showing the predicted protein structure of RBM47, its closest homologue A1CF, a known RNA-binding protein, and the predicted structure of RBM47[I281fs] mutant. Blue diamonds represent RRM motifs, pink rectangle represents a truncated RRM motif.

The following figure supplements are available for figure 1:

**Figure supplement 1**. *RBM47* expression and genetic alterations in human breast cancer.

*Wang et al., 2005*) with mutational data from a brain metastasis that originated from a basal breast cancer (*Ding et al., 2010*; *Figure 1A*). Specifically, we looked for genes that had reduced mRNA expression in functionally metastatic cancer cells, evidence for low mRNA expression associated with poor patient outcome in clinical samples, and an enriched mutation in the brain metastasis sequenced by *Ding et al. (2010)*. *RBM47*, a gene encoding a previously uncharacterized putative RNA-binding protein was the only one that fulfilled all these criteria (*Baltz et al., 2012*; *Castello et al., 2012*; *Ray et al., 2013*). We confirmed the lower expression of *RBM47* mRNA in the highly metastatic cells (*Figure 1B*). This translated into a comparable difference at the protein level (*Figure 1C*). In the clinical data sets, low *RBM47* mRNA expression was significantly associated with relapse to brain and lung (*Figure 1D,E*) but not to bone (*Figure 1F*). In multivariate analysis combining *RBM47* expression with estrogen, progesterone and HER2 receptor status (ER, PR and HER2), the association with brain metastasis remained statistically significant (*Figure 1—figure supplement 1A*). We further characterized the expression patterns of *RBM47* in the TCGA cohort of 748 breast cancer samples studied by RNA-seq (*Cerami et al., 2012*; *Cancer Genome Atlas Network, 2012*). We found that low *RBM47* expression was significantly associated with claudin-low and basal breast cancers (*Figure 1G*), two subtypes of poor prognosis (*Smid et al., 2008*; *Lu et al., 2013*).

The *RBM47*[I281fs] mutation first reported in a brain metastasis truncates the protein from the third RNA recognition motif (RRM) onwards (*Figure 1H*). As this mutation was already present in a minority subpopulation of the corresponding primary tumor (*Ding et al., 2010*), we looked for additional evidence of genetic *RBM47* aberrations in primary breast cancer cohorts. The catalogue of somatic mutations in cancer (COSMIC) database (*Forbes et al., 2010*) reported 9 non-synonymous mutations in *RBM47*, three of which were frameshift mutations truncating one or more of the RRM domains (*Figure 1—figure supplement 1B*). Furthermore, analysis of the data from the TCGA cohort revealed that in basal breast cancer, *RBM47* was targeted by a mutation or homozygous deletion in ~8% of the cases (*Figure 1—figure supplement 1C*). Moreover, heterozygous loss of the *RBM47* locus was present in 30% of the TCGA cohort (*Cerami et al., 2012*). These correlative analyses of multiple different breast cancer data sets, both experimental systems and large clinical patient cohorts, suggested that reduced expression or function of RBM47 is associated with breast cancer progression already within primary tumors, and that clones with reduced RBM47 function may display enhanced lung and brain metastatic fitness.

## RBM47 suppresses breast cancer progression

In order to test the role of *RBM47* as a suppressor of breast cancer progression, we stably introduced both wild type *RBM47* and *RBM47*[I281fs] in the lung metastatic (LM2) and brain metastatic (BrM2) derivatives of the MDA-MB-231 triple negative breast cancer cells (MDA231 for short), respectively (*Figure 2—figure supplement 1A*; *Minn et al., 2005*; *Bos et al., 2009*). Of note, despite robust mRNA expression, the mutant RBM47[I281fs] protein levels were low, indicating that the mutant protein was unstable and therefore inactive (*Figure 2—figure supplement 1B*). As determined by in vivo bioluminescence in experimental metastasis assays, wild type *RBM47* inhibited lung colonization by the MDA231-LM2 cells, when compared to *RBM47*[I281fs] (*Figure 2A–B*). Similarly, we observed extended brain metastasis free survival in mice inoculated with the MDA231-BrM2 cells expressing *RBM47* when compared to those expressing *RBM47*[I281fs] (*Figure 2C*). This difference translated into reduced brain metastatic burden as determined by ex vivo imaging (*Figure 2D,E*, *Figure 2—figure supplement 1C*). We then tested the effects of inhibiting endogenous RBM47 in the metastatic colonization of the parental MDA231 cells. With comparable effects of RBM47 re-expression seen in both the lung and brain metastasis models (*Figure 2A,E*), we chose the lung colonization assay for loss-of-function studies as this allowed the simultaneous analysis of a greater number of tumor re-initiation events. Two *RBM47*-targeting shRNA constructs significantly shortened the lung metastasis free survival in mice when compared to controls (*Figure 2F*, *Figure 2—figure supplement 1D*), as determined by bioluminescence imaging (*Figure 2G,H*). This result was confirmed with a second more indolent cancer cell clone (*Figure 2—figure supplement 1E–G*).

## Clonal heterogeneity in RBM47 sensitivity

The initial functional experiments suggested that the tumor suppressive effect of RBM47 on the overall population of metastatic cancer cells was modest. This could reflect either weak tumor

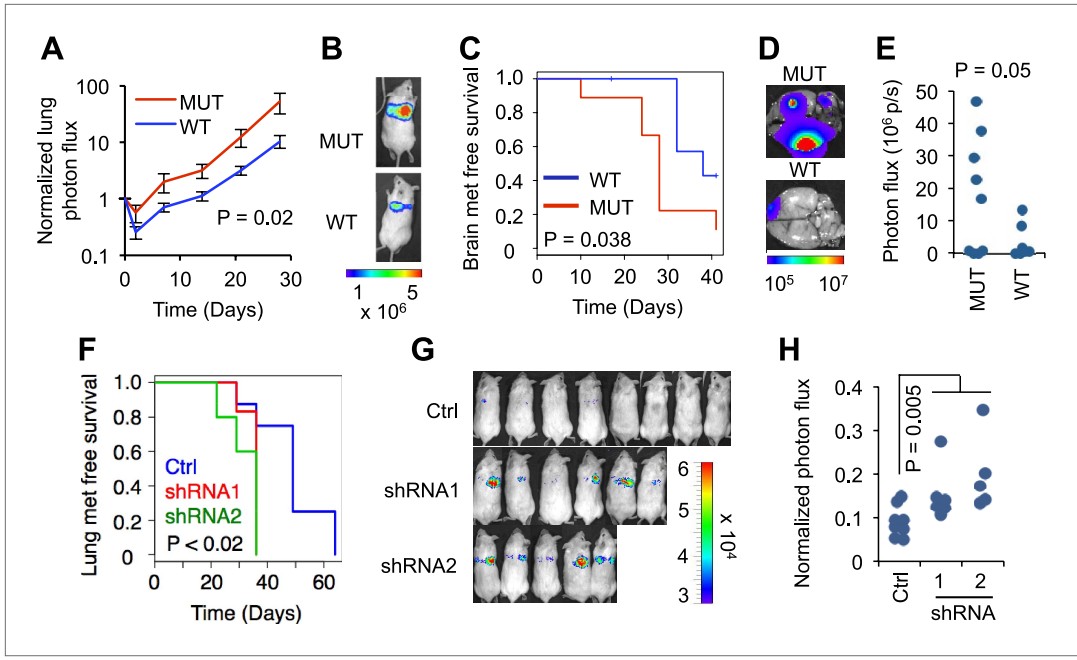

**Figure 2**. RBM47 suppresses metastatic breast cancer progression. (**A**) Normalized lung photon flux in mice after tail vein inoculation of MDA231-LM2 cells expressing either wild type *RBM47* or *RBM47^I281fs*. p-value calculated utilizing repeated measures two-way ANOVA. N = 6 for *RBM47^I281fs*, N = 8 for *RBM47*. (**B**) Representative biolumi-nescence images from the experiment shown in (**A**). The color scale shows bioluminescence (photons/second). (**C**) Brain metastasis free survival as determined by in vivo bioluminescence imaging in mice after intracardiac inoculation of MDA231-BrM2 cells expressing either wild type *RBM47* or *RBM47^I281fs*. p-value calculated using the log-rank test. N = 9 for *RBM47^I281fs*, N = 7 for *RBM47*. (**D**) Representative bioluminescence images from the experiment shown in (**C**). The color scale shows bioluminescence (photons/second). (**E**) Ex vivo quantification of bioluminescence from brain metastases on day 42 of the experiment shown in panel (**C**). P-value calculated by two-tailed Student's *t* test. (**F**) Lung metastasis free survival as determined by in vivo bioluminescence imaging in mice after tail vein inoculation of parental MDA231 cells expressing either control vector (pGIPZ) or hairpins against RBM47 (shRNA1 and shRNA2). p-values calculated using the log-rank test. N = 8 for Ctrl group, N = 6 for shRNA1, N = 5 for shRNA2. (**G**) In vivo bioluminescence imaging on day 36 of the experiment shown in (**F**) demonstrates earlier emergence of detectable lung metastasis for the RBM47 knockdown groups when compared to the control animals. The color scale shows bioluminescence (photons/second). (**H**) Quantification of bioluminescence for the time point shown in (**G**). Data normalized to signal on day 0 for each animal. p-value calculated using the Wilcoxon rank-sum test.

The following figure supplements are available for figure 2:

**Figure supplement 1**. RBM47 suppresses breast cancer progression.

suppressive function of RBM47 in general, heterogeneity in the sensitivity to RBM47 among dif-ferent cancer cell subpopulations, or in the case of RBM47 reintroduction, loss of transgene expres-sion. In order to distinguish between these possibilities, we used immunohistochemistry to assess RBM47 expression in the lung nodules formed either by wild type *RBM47* or *RBM47^I281fs* express-ing cells. This revealed that some cancer clones were able to form robust lung metastasis even in the presence of RBM47 (**Figure 3A**, **Figure 3—figure supplement 1A**), but that many of the metastases formed after the inoculation of wild type RBM47-expressing cells had avoided or sup-pressed the expression of RBM47 (**Figure 3B**, **Figure 3A—figure supplement 1A**). As expected, the *RBM47^I281fs*-expressing tumors contained only weakly staining cancer cells intermingled with small cells with strong RBM47 expression (**Figure 3C**), similar to those seen in normal lung paren-chyma (**Figure 3D**). The rate of proliferation as determined by Ki67 immunohistochemistry did not correlate with the level of RBM47 expression (**Figure 3—figure supplement 1A**). This was in line with the idea that some clones were more sensitive to RBM47 than others, but also that a strong

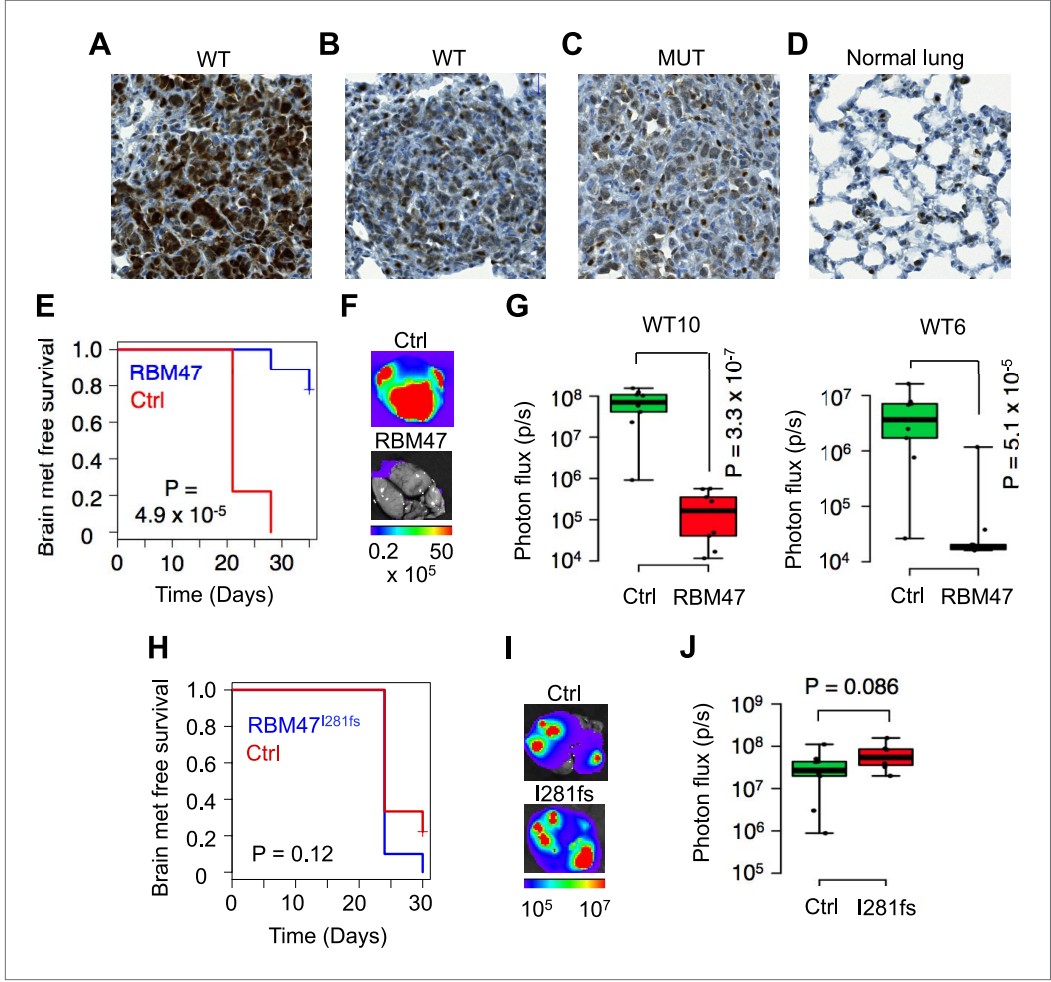

**Figure 3**. Clonal heterogeneity in RBM47 sensitivity. (**A**) A lung metastatic nodule with strong RBM47 expression in a mouse inoculated with RBM47-transduced MDA231-LM2 cells. RBM47 protein expression detected by immuno-histochemistry using an antibody against RBM47. (**B**) A lung metastatic nodule with weak RBM47 expression in a mouse inoculated with RBM47-transduced MDA231-LM2 cells. RBM47 protein expression detected by immunohis-tochemistry. Note the clearly reduced staining when compared to panel (**A**). (**C**) A lung metastatic nodule with weak RBM47 expression in a mouse inoculated with *RBM47^I280fs*-transduced MDA231-LM2 cells. RBM47 protein expression detected by immunohistochemistry. Staining intensity similar to that seen in panel (**B**). (**D**) RBM47 expression in normal mouse lung. RBM47 protein expression detected by immunohistochemistry. Note small cells with strong RBM47 expression, similar to those seen in panels (**B**) and (**C**). (**E**) Brain metastasis free survival as determined by in vivo bioluminescence imaging in mice after intracardiac inoculation of WT10 cells. The RBM47 group received doxycycline in diet. p-value calculated using the log-rank test. N = 9 for both groups. (**F**) Representative ex vivo bioluminescence images from brain metastasis of the experiment shown in (**E**). The color scale shows biolumines-cence (photons/second). (**G**) Ex vivo quantification of bioluminescence from brain metastases with and without RBM47 reintroduction. WT10 data from the experiment shown in (**E**). WT6 data from a similar experimental setup. p-value calculated by two-tailed Student's *t* test. N = 9 for Ctrl group, N = 10 for RBM47 group. (**H**) Brain metastasis free survival as determined by in vivo bioluminescence imaging in mice after intracardiac inoculation of MUT3 cells. The I281fs group received doxycycline in diet. p-value calculated using the log-rank test. N = 9 for Ctrl group, N = 7 for RBM47^I281fs group. (**I**) Representative ex vivo bioluminescence images from brain metastases of the experiment shown in (**H**). The color scale shows bioluminescence (photons/second). (**J**) Ex vivo quantification of biolumines-cence from brain metastases of the experiment shown in (**H**). p-value calculated by two-tailed Student's *t* test.

The following figure supplements are available for figure 3:

**Figure supplement 1**. RBM47 expression in experimental metastases.

**Figure supplement 2**. Model systems with inducible RBM47 expression.

selective pressure led metastatic cells to lose RBM47, a finding consistent with the initial observation of RBM47 loss in metastatic cell populations (*Figure 1C*).

In order to allow better experimental control over RBM47 expression we utilized a conditional expression system. By focusing on the two brain metastatic cell lines that expressed the lowest levels of endogenous RBM47 (*Figure 1C*) we generated single cell-derived clones with doxycycline-inducible expression of either wild type RBM47 (henceforth WT10 and WT6, respectively) or the patient-derived mutant RBM47[I281fs] (henceforth MUT3). All clones exhibited dose-dependent RBM47 mRNA upregulation upon doxycycline induction (*Figure 3—figure supplement 2A–C*) that translated into increased RBM47 protein expression in the wild type-expressing clones (*Figure 3—figure supplement 2D,E*). Cancer cells also tolerated RBM47 in vitro (*Figure 3—figure supplement 2F*). After confirming the feasibility of doxycycline-mediated conditional gene activation in metastatic brain lesions (*Figure 3—figure supplement 2G*), we inoculated WT6, WT10 and MUT3 cells into immunocompromized mice and assessed their brain-metastatic phenotype. All clones formed robust brain metastases under doxycycline-free conditions (*Figure 3E–J*). The induction of *RBM47* expression with doxycycline in the diet inhibited robustly brain colonization of both metastatic cell clones, WT6 and WT10 (*Figure 3E–G*). However, the patient-derived mutant *RBM47[I281fs]* did not show any tumor suppressive effects (*Figure 3H–J*). Collectively, these results demonstrated that RBM47 was able to strongly inhibit metastatic functions of some cancer clones, whereas other clones were able to form metastasis despite the presence of RBM47.

## Transcriptome-wide identification of RBM47 binding sites

To determine whether RBM47 can directly bind RNA in vivo, we made use of the ability of UV-irradiation at 254 nm to induce chemical crosslinks between RNA and proteins that are in direct contact (*Darnell, 2010*). γ-$^{32}$P-labeled RNA was detected by autoradiogram at ~76 kD, the predicted size of Flag-RBM47, in Flag-immunoprecipitates from UV-irradiated, doxycycline-treated MDA231-BrM2 WT Flag-RBM47 cells, but not doxycycline-treated empty vector control or non-irradiated WT Flag-RBM47 expressing cells (*Figure 4A*). To identify directly bound RBM47 targets, a modified version of the high throughput sequencing and cross-linking immunoprecipitation (HITS-CLIP) protocol (*Licatalosi et al., 2008*; *Weyn-Vanhentenryck et al., 2014*) was carried out in duplicate on Flag-RBM47 expressing MDA231-BrM2 cells treated with doxycycline. (*Figure 4B*, see 'Materials and methods'). RBM47-bound HITS-CLIP reads were mapped to the human genome, yielding ~7.7 × 10$^6$ and ~2.0 × 10$^6$ unique reads (tags) per replicate. 75% of the tags mapped to regions corresponding to UCSC/Refseq genes, with a high degree of reproducibility of binding observed between replicates (Spearmann correlation coefficient $R^2$ = 0.998, total tags per gene, *Figure 4C*).

To identify robust and reproducible RBM47-binding sites, tags were clustered to return regions with evidence of binding in both replicates (biological complexity, BC = 2) with increasingly stringent filters (*Figure 4D*): tags per cluster (tags ≥2, 617,026 clusters; tags ≥10, 94,966 clusters), a previously described significant peak threshold (significant peak height ≥10, 29,562 clusters [*Chi et al., 2009*]), and a ranked reproducibility chi-squared score (p≤0.01, 19,433 clusters [*Darnell et al., 2011*]). As has been previously described for the n-ELAV proteins (*Ince-Dunn et al., 2012*), identification of the most robust binding through increased stringency of cluster definition led to an increase in proportional binding in 3′UTR regions. For RBM47 this occurred due to loss of the majority of reproducible, yet relatively small intronic binding sites (*Figure 4E*), which may reflect the relative abundance of pre- and mature RNA message. Motif analysis (MEME, *Bailey and Elkan, 1994*) failed to identify an enriched RBM47-binding sequence in ±10 nt footprints centered on the top 3000 significant peaks (data not shown), but revealed a polyU sequence enriched in RBM47-binding sites containing cross-link induced mutations (deletion CIMS, *Zhang and Darnell, 2011*, *Figure 4F*). The apparent lack of an enriched RBM47-binding motif within robust CLIP-derived binding sites may reflect the broad binding patterns observed, as exemplified by the predominantly 3′UTR binding in *DKK1* and *IL8* (*Figure 4G*).

## RBM47 regulates alternative splicing

Reproducible binding of RBPs in intronic regions has proven to be predictive of a role in pre-mRNA processing for multiple proteins. To explore the relationship between RBM47 intronic binding and alternative splicing, RNA-seq was carried out in triplicate to compare MDA231-BrM2 cells and

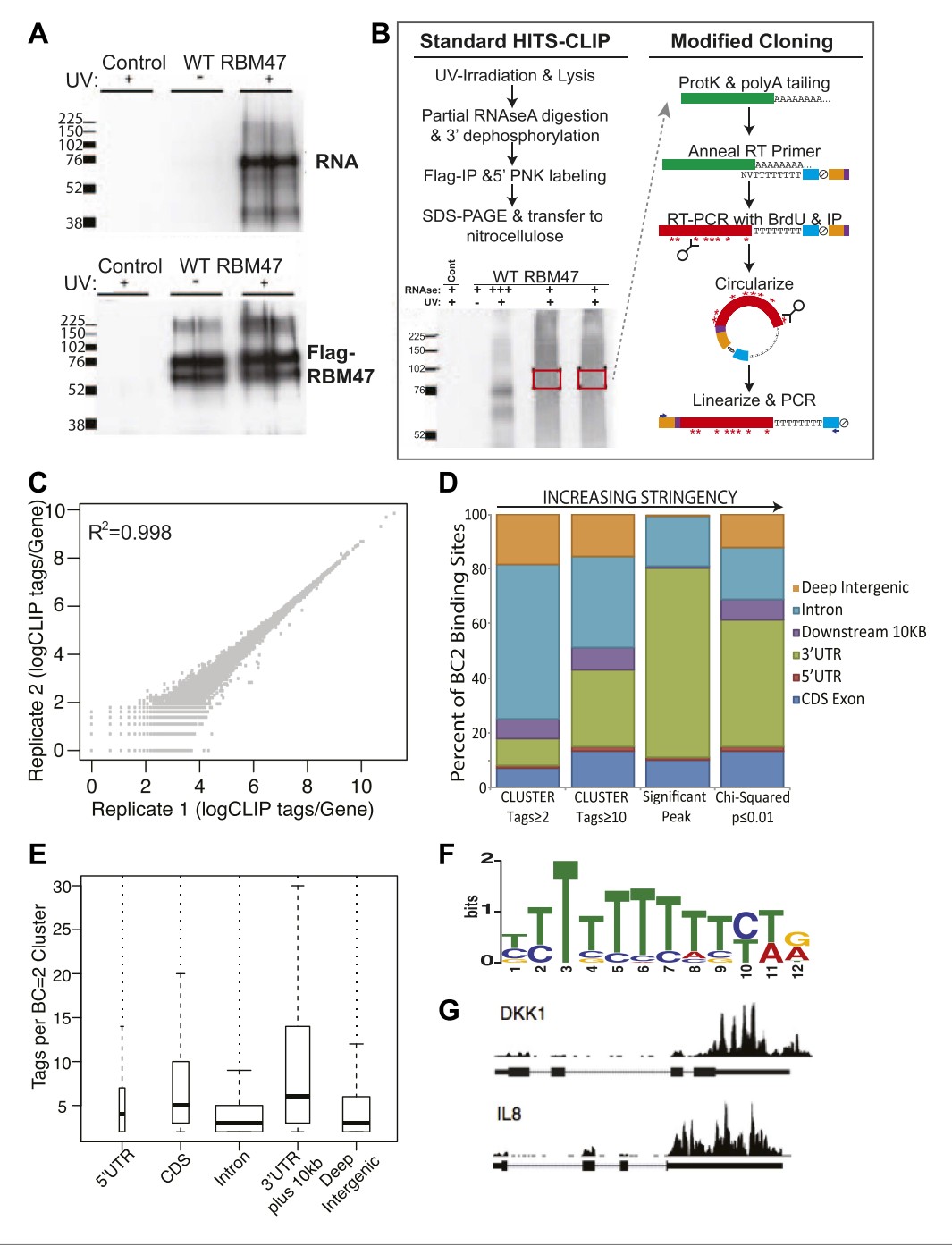

**Figure 4**. HITS-CLIP identifies genome-wide RBM47 binding maps. (**A**) Radiolabelled RNA is detectable in RBM47-expressing 231-BrM2 metastatic cells that have been UV-irradiated indicating in vivo RNA binding ability. No RNA is detected in non-crosslinked cells despite the presence of ample immunoprecipitated Flag-RBM47 protein. No RNA or protein is detected in control 231-BrM2 transduced with empty vector. Samples run in duplicate. (**B**) Schematic of the modified HITS-CLIP protocol showing autoradiogram of duplicate Flag-RBM47 samples used. Purified RBM47-bound RNA fragments (green) were polyA tailed and reverse transcribed in the presence of Brd-U using a polydT-NV primer encoding the full sense sequence of the Illumina reverse sequencing primer (blue), an abasic furan that serves as an ApeI cut site (ξ), a partial reverse complement to the Illumina forward sequencing primer (orange), and two hexamer sequences (purple): a known-sequence index for multiplexing and a degenerate barcode used to distinguish unique cDNA clones from PCR duplicates. cDNA were

*Figure 4. Continued on next page*

*Figure 4. Continued*

stringently purified, circularized and linearized using ApeI to bring the Illumina sequence into correct orientation with respect to the cloned fragment, and the samples PCR amplified and deep sequenced. (**C**) RBM47 HITS-CLIP is highly reproducible between replicate experiments at the level of unique CLIP tags per transcript. (**D**) Increasing the stringency of biologically reproducible RBM47 binding site definition reveals predominant binding in 3'UTRs and intronic regions of target transcripts, with the most robust binding (tags per binding site) evident in 3'UTRs. (**E**) Distribution of tags number per biologically reproducible cluster in coding and non-coding regions of RBM47-targeted transcripts reveals a bimodal binding pattern between 3'UTRs and introns, with the latter having large numbers of reproducible yet less robust binding. (**F**) MEME analysis reveals an enrichment for polyU sequences (50 sites, p=$2.4e^{-16}$) in the ±10 nt foot print region surrounding reproducible RBM47 deletion CIMS (357 sites with ≥5 mutations, FDR ≤0.01, *Zhang and Darnell, 2011*). (**G**) Widespread RBM47 binding is seen in target transcripts, as exemplified by binding patterns seen in the 3'UTRs of DKK1 and IL8.

RBM47-expressing WT10 cells. Reads were mapped to cassette (CA) exon junctions as described previously for Mbnl2 (*Charizanis et al., 2012*), and an average inclusion rate (IR) calculated for each cell type to allow for the identification of reciprocal splicing changes while normalizing for changes in RNA stability (*Ule et al., 2005*; *Figure 5A*). The average change in inclusion rate (ΔI) was then calculated such that positive ΔI indicates RBM47-dependent cassette exon inclusion. This analysis revealed 121 and 140 CA exons with significant RBM47-dependent inclusion and exclusion, respectively. To assess whether RBM47-binding occurred in the vicinity of these splice sites, HITS-CLIP tags in BC2 tags ≥5 clusters (to account for lower levels of intronic binding seen in *Figure 4E*) in the region of the alternative splice were calculated. Forty-eight RBM47-bound included and 49 RBM47-bound excluded CA exons were identified in a total of 84 genes (*Figure 5B*; *Supplementary file 1*). RBM47-dependent splicing changes were confirmed via RT-PCR as shown for *SLK*, *MDM4*, *LIMCH1*, *MBNL1* and *SEC31A* (*Figure 5C*, *Figure 5—figure supplement 1*).

By mapping normalized RBM47 CLIP tags associated with RBM47-dependent splicing changes on a composite transcript (*Licatalosi et al., 2008*) we generated an RNA binding map of consensus binding sites within 1 kb of exon-intron boundaries, with respect to exon inclusion or exclusion (*Figure 5D*). The resulting normalized complexity map reveals enriched RBM47 binding in the vicinity of splice acceptor sites of the CA exon and 3' CE for included alternative isoforms, while relative enrichment of RBM47 binding was seen at the 5' CE splice donor and 3'CE splice acceptor of excluded isoforms.

## RBM47 affects mRNA steady state levels

To further study the functional consequences of RBM47-mRNA binding events, we determined global steady state mRNA levels in the RBM47 expressing brain metastatic cells by RNA-seq. We took advantage of the clonal doxycycline-inducible cell line systems, which facilitated the analysis of RBM47 dose-dependence. First, genome-wide analyses showed that the mRNA level of several more genes correlated significantly with increasing concentrations of doxycycline in both cell lines expressing the wild type RBM47 (WT6 and WT10), when compared to cells expressing the mutant (*Figure 6A*). This result indicated that RBM47 elicits dose-dependent changes in mRNA levels. The correlation coefficients followed a pattern that suggested the existence of mRNA species that correlated both positively and negatively with wild type, but not mutant, RBM47 expression, that is few genes in MUT3 cells had correlation coefficients close to 1 or −1, whereas in WT6 and WT10 cells such genes were abundant (*Figure 6B*). Encouraged by these observations, we looked for mRNAs that fulfilled the following criteria: (i) p-value of correlation less than 0.01 in both WT6 and WT10 cells, (ii) mRNA expression change detectable already at the lowest levels of RBM47 expression in both cell clones and (iii) no significant correlation with RBM47$^{I281fs}$ expression. This revealed a set of 102 mRNAs that were upregulated and 92 that were downregulated, respectively, in cells expressing the wild type RBM47 (*Figure 6C*, *Figure 6—figure supplement 1A,B*; *Supplementary file 2*). Importantly, these changes were observed already with the lowest expression level of RBM47 that was comparable or lower than those detected in endogenously RBM47 expressing cells (*Figure 3—figure supplement 2D,E*).

To determine whether the 194 RBM47-responsive genes displayed clinically meaningful expression patterns, we conducted an unsupervised hierarchical clustering analysis of the TCGA cohort of breast

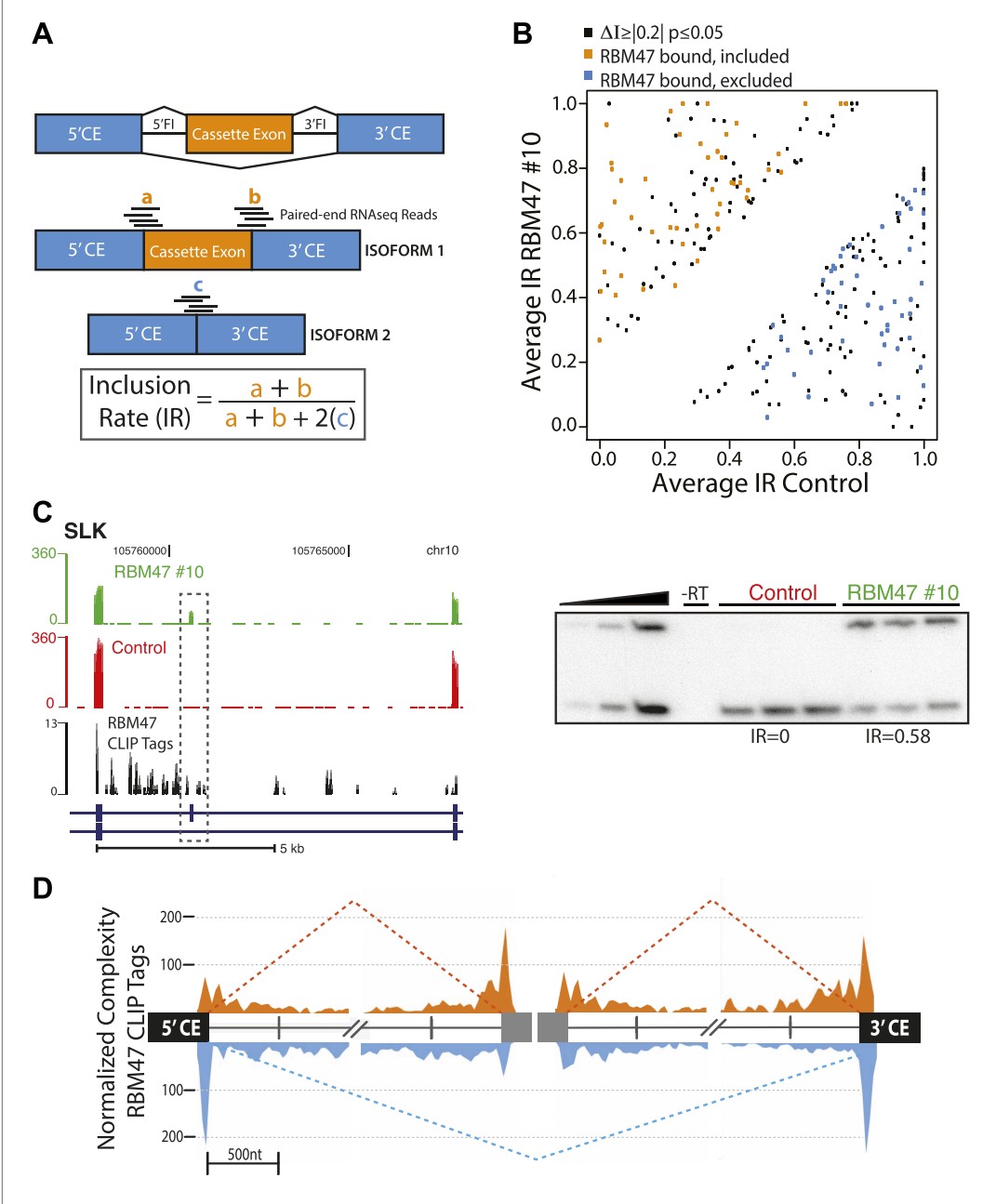

**Figure 5**. RBM47 regulates alternative splicing. (**A**) Schematic showing the method used to calculate alternative exon inclusion rates from paired-end RNA-seq data. 5'CE–5' flanking constitutive exon, 3'CE–3' flanking constitutive exon, 5'FI–5' Flanking intron, 3'FI–3' Flanking intron. (**B**) Scatter plot of all expressed alternatively spliced CA exons showing RBM47-dependent change in inclusion (black, ≥10 RNA-seq reads spanning exon–exon junctions, ΔI ≥|0.2|, p≤0.05) with orange points indicating RBM47-bound and included CA exons, and blue points indicating bound and excluded exons, respectively. CA exons were considered bound given a total of tags ≥10 in BC2 tags ≥5 clusters mapping to the region spanning the start of the 5'CE to the end of the 3'CE. p-values calculated by Fisher's exact test using total isoform 1 and total isoform 2 RNA-seq reads in each condition. (**C**) Left panel shows a section of the SLK transcript (blue) that includes a CA exon (grey box). The top two panels show RNA-seq data from WT10 (green) and control cells (red), with RBM47 HITS-CLIP tags mapping to the region shown beneath in black. Increased RNA-seq signal corresponding to the CA exon is seen in the presence of RBM47 expression, while robust binding is evident in the 5'FI. Independent RT-PCR validation of this splice is shown in the right panel, with IR calculated using ImageJ analysis of autoradiograms. (**D**) Normalized complexity map of RBM47-dependent alternative splicing

*Figure 5. Continued on next page*

*Figure 5. Continued*

of CA exons. Orange and blue peaks represent binding associated with RBM47-dependent exon inclusion and exclusion, respectively.

The following figure supplements are available for figure 5:

**Figure supplement 1**. RBM47-dependent splicing events.

cancer specimens. Two main clusters emerged, one of which harbored characteristics of the RBM47-low phenotype ('Cluster 2', *Figure 6—figure supplement 1C*). First, this subgroup, Cluster 2, had significantly lower RBM47 expression levels when compared to Cluster 1 (*Figure 6D*). Second, the genes that were induced upon RBM47 reintroduction tended to have lower expression in Cluster 2, whereas genes with reduced expression level in the RBM47-expressing cells tended to show higher expression in Cluster 2 (*Figure 6E*). These observations validated in clinical tumor samples the RBM47-dependent gene expression correlations identified in vitro.

To identify directly regulated targets of RBM47, we combined the mRNA expression data with the HITS-CLIP-derived RBM47 transcriptome-wide binding data. Of the 2498 strongest binding partners with >100 tags per transcript (total tags in BC2 tags ≥10 clusters; *Supplementary file 3*), 25 were among the 102 RBM47-upregulated genes and 17 among the 92 RBM47-downregulated genes (*Figure 6F*), indicating no significant binding preference for either groups. This was reflected in the similar overall RBM47 binding profiles in both the up- and down-regulated genes (*Figure 6G*). Two of the top-scoring RBM47 binding mRNAs, *IL8* and *DKK1* (17,079 and 16,208 tags in clusters per gene, respectively, *Figure 4G*), were among the upregulated genes. This increase in mRNA levels was associated with increased protein secretion as determined by ELISA, whereas VEGFA, the mRNA of which was bound but not upregulated by RBM47, showed no change in protein secretion (*Figure 6H*).

## RBM47 modulates *DKK1* mRNA stability

Nuclear RNA binding proteins typically function in large multiprotein complexes that regulate mRNA biogenesis (*Dreyfuss et al., 2002*). Our data from both genome-wide HITS-CLIP and RNA-seq analysis was compatible with RBM47 being a member of these RNA chaperone units. This suggested that RBM47 may not necessarily have a direct tumor suppressive signaling function. Rather, it raised the possibility that loss of RBM47, leading to subtle changes in multiple mRNAs, either through stabilization, destabilization or alternative splicing, could be selected for if the net effect of both growth-promoting and growth-inhibiting changes would be beneficial for cancer cells under the stress of dissemination to and colonizing distant organs. In line with this, both the up- and down-regulated target genes of RBM47, as well as the genes that were targets of RBM47-mediated alternative splicing, contain genes that have previously been shown to either promote or inhibit tumor phenotypes (*Supplementary file 1* and *Supplementary file 3*). For example, *DKK1* (*Bafico et al., 2004*; *Cowling et al., 2007*; *Mikheev et al., 2008*), *HTATIP2* (*Zhao et al., 2007*), *HBP1* (*Paulson et al., 2007*; *Li et al., 2011*), *MXI1* (*Lahoz et al., 1994*) and *CASP7* (*Hudson et al., 2013*), all bound by RBM47 and upregulated upon RBM47 reintroduction, have known tumor suppressive functions. Similarly, RBM47-induced splicing changes were seen in genes such as *SLK* (*Roovers et al., 2009*), *MDM4* (*Wade et al., 2013*) and *TNC* (*Oskarsson et al., 2011*), all of which are genes with known functions in cancer.

Focusing on *DKK1*, one of the most robustly bound RBM47 target transcripts identified by HITS-CLIP, we investigated the possible role of RBM47 as a modulator of mRNA abundance. As predicted by our results in WT6 and WT10 cells, knockdown of RBM47 in two additional breast cancer cell lines expressing high levels of endogenous RBM47, SKBR3 and ZR-75-30, reduced *DKK1* mRNA levels (*Figure 7A*, *Figure 7—figure supplement 1A*). This validated RBM47 as a modulator of *DKK1* mRNA level in breast cancer cells.

The fact that RBM47 bound to *DKK1* mRNA 3'UTR and increased *DKK1* mRNA levels suggested the possibility that RBM47 had the capability of stabilizing *DKK1* mRNA. We tested this by treating cancer cells with actinomycin D, a general inhibitor of transcription, and measuring *DKK1* mRNA levels in the following hours. This demonstrated that wild type *RBM47*, but not the *RBM47*[I281fs] mutant, was able to increase the half-life of *DKK1* mRNA by up to fourfold (*Figure 7B*).

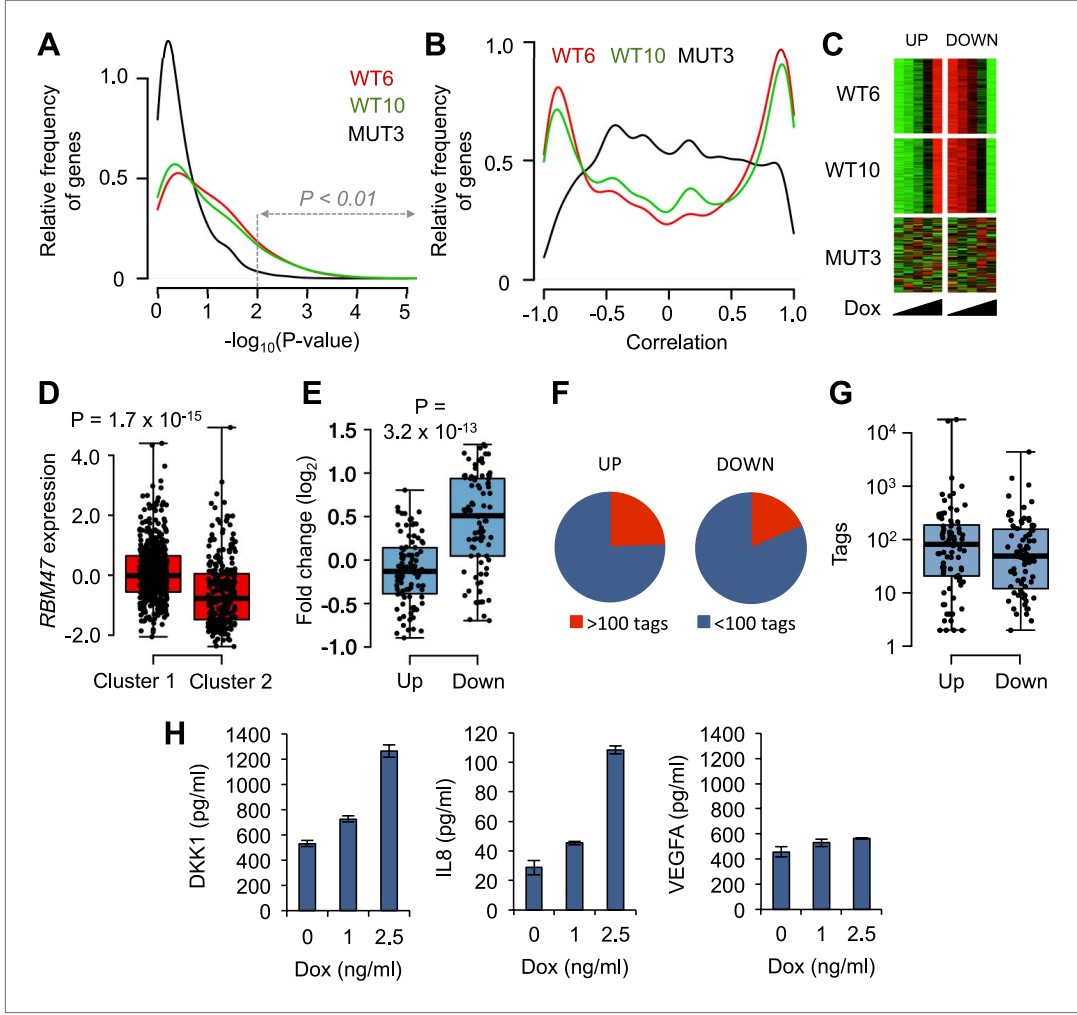

**Figure 6**. RBM47-induced changes in mRNA levels. (**A**) Distribution of p-values from correlation analysis of doxycycline concentration and gene expression for all genes in WT6, WT10 and MUT3 cell lines, respectively. Global gene expression determined by RNA-seq. (**B**) Distribution of correlation coefficients between doxycycline concentration and gene expression in WT6, WT10 and MUT3 cell lines, respectively. (**C**) Heat maps showing the top 102 positively (UP) correlated and 92 negatively (DOWN) correlated genes with RBM47 expression in WT6 and WT10 cells. The expression of these genes does not correlate with *RBM47[I280fs]* expression in the MUT3 cells. (**D**) *RBM47* mRNA expression in the TCGA cohort of breast cancer samples classified by the clusters shown in *Figure 6—figure supplement 1C*. p-value determined by two-tailed Student's *t* test. (**E**) Fold change between Cluster 1 and Cluster 2 shown for the 102 positively and 92 negatively correlated RBM47 target genes, respectively. Positive fold change shows higher expression in Cluster 2, which has lower expression of RBM47. The genes that are induced upon RBM47 reintroduction tend to have lower expression in Cluster 2, and the genes that show lower expression upon RBM47 reintroduction tend to have higher expression in Cluster 2. This is in line with the experimental results shown in panel (**C**). P-value determined by two-tailed Student's *t* test. (**F**) Pie charts showing the fraction of target genes with more than 100 RBM47 tags for both the 102 positively and 92 negatively correlated RBM47 target genes. (**G**) Tags per transcript plotted for both the positively and negatively correlated RBM47 target genes that showed more than 1 tag. The only two binding partners with >$10^4$ tags represent *DKK1* and *IL8*, respectively. (**H**) Secreted DKK1 and IL8 protein levels determined by ELISA in WT6 cells treated with the indicated doxycycline concentrations. VEGFA used as a control.

The following figure supplements are available for figure 6:

**Figure supplement 1**. Transcriptomic signature of RBM47 reintroduction.

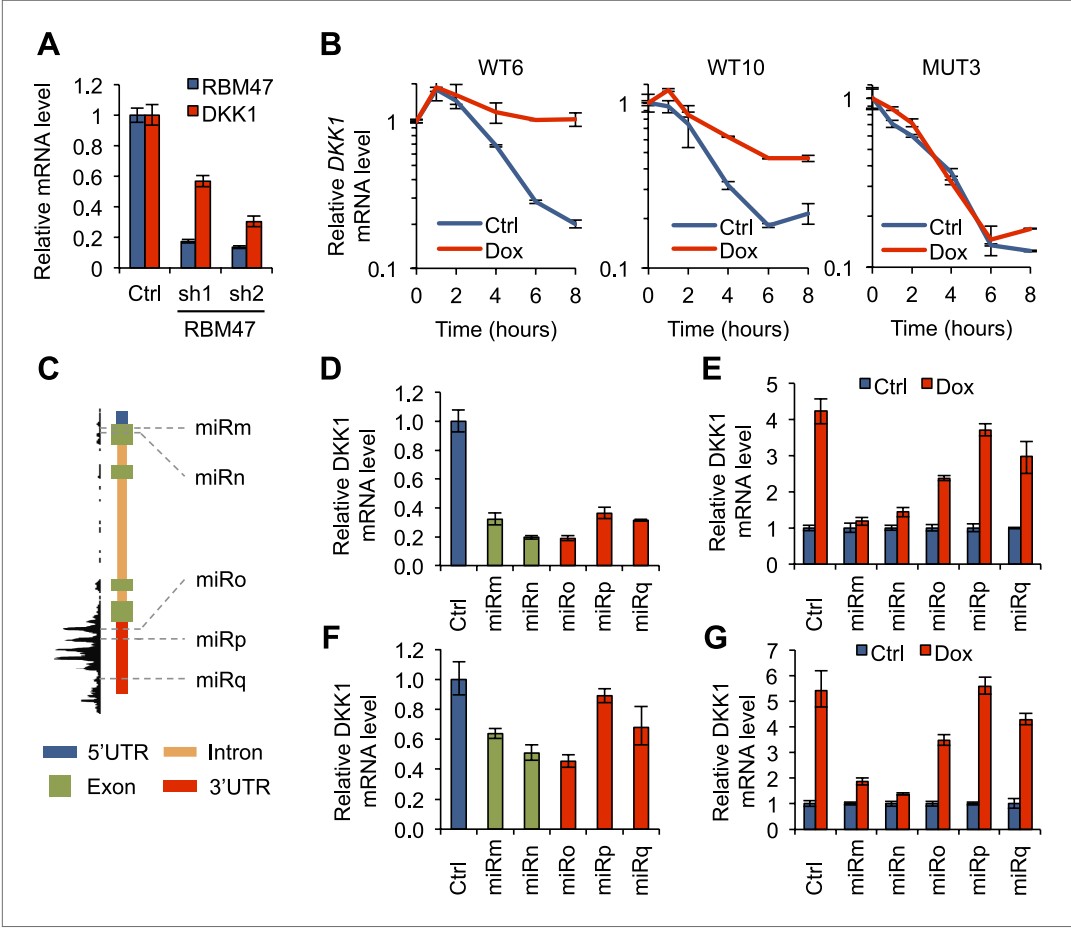

**Figure 7**. RBM47 modulates *DKK1* mRNA stability. (**A**) *RBM47* and *DKK1* mRNA expression measured by quantitative real-time RT-PCR in SKBR3 cells expressing either control vector (pGIPZ) or hairpins against *RBM47* (shRNA1 and shRNA2). Error bars represent 95% confidence intervals obtained from multiple PCR reactions. (**B**) *DKK1* mRNA stability determined by measuring mRNA levels by quantitative real-time RT-PCR in WT6, WT10 and MUT3 cells, treated with or without doxycycline, after inhibition of transcription with actinomycin D. Data normalized to time point 0. Error bars represent 95% confidence intervals obtained from multiple PCR reactions. WT6: $T_{1/2}$ Ctrl = 2.3 hr, $T_{1/2}$ Dox = 9.8 hr; WT10: $T_{1/2}$ Ctrl = 2.4 hr, $T_{1/2}$ Dox = 5.4 hr; MUT3: $T_{1/2}$ Ctrl = 2.1 hr, $T_{1/2}$ Dox = 2.2 hr. (**C**) Schematic showing the locations of different DKK1 miR-constructs in relation to *DKK1* genomic locus and RBM47 binding patterns. miRm and miRn target exon 1 that is not bound by RBM47. miRo, miRp and miRq target *DKK1* 3′UTR that is strongly bound by RBM47. (**D**) *DKK1* mRNA expression measured by quantitative real-time RT-PCR in WT6 cells expressing either control vector (pGIPZ) or the five *DKK1*-targeting miR constructs shown in panel (**C**). Error bars represent 95% confidence intervals obtained from multiple PCR reactions. (**E**) *DKK1* mRNA expression measured by quantitative real-time RT-PCR in WT6 cells expressing either control vector (pGIPZ) or the five *DKK1*-targeting miR constructs, with or without doxycycline treatment. Data normalized to the non-treated control for each cell line separately. Error bars represent 95% confidence intervals obtained from multiple PCR reactions. (**F**) *DKK1* mRNA expression measured by quantitative real-time RT-PCR in WT10 cells expressing either control vector (pGIPZ) or the five *DKK1*-targeting miR constructs shown in panel (**C**). Error bars represent 95% confidence intervals obtained from multiple PCR reactions. (**G**) *DKK1* mRNA expression measured by quantitative real-time RT-PCR in WT10 cells expressing either control vector (pGIPZ) or the five *DKK1*-targeting miR constructs, with or without doxycycline treatment. Data normalized to the non-treated control for each cell line separately. Error bars represent 95% confidence intervals obtained from multiple PCR reactions.

The following figure supplements are available for figure 7:

**Figure supplement 1**. Effects of RBM47 knockdown on DKK1 mRNA.

## RBM47 protects *DKK1* 3′UTR from destabilizing factors

As RBM47 binding to *DKK1* was concentrated on the 3′UTR, a known region of regulatory activity (*Zhou et al., 2012*), we considered the possibility that RBM47 could compete with microRNAs or other mRNA destabilizing factors that target the 3′UTR (*Bhattacharyya et al., 2006*; *Young et al., 2012*). To test this experimentally, we generated miR-30-based shRNA-miR constructs that targeted different regions of the *DKK1* transcript, two at the 5′ end with minimal RBM47 binding and three at the 3′ end with abundant RBM47 signal (*Figure 7C*; *Fellmann et al., 2011*). All constructs knocked the *DKK1* transcript level down by 65–80% in the WT6 cells when no doxycycline was present (*Figure 7D*). The induction of RBM47 expression did not have a significant effect on the efficiency on the 5′-targeting shRNA-miRs (*Figure 7E*). In contrast, RBM47 inhibited the capability of the 3′-targeting shRNA-miRs to keep the *DKK1* mRNA level down (*Figure 7E*). Similar observations were made with the WT10 cells (*Figure 7F,G*). Argonaute HITS-CLIP in the MDA231-BrM2 cells (CBM et al., unpublished data) indicates robust binding to the DKK1 3′UTR, supporting the regulatory role of this region. We conclude that the effects of RBM47 on *DKK1* mRNA levels may be due to the ability of RBM47 to protect this mRNA from destabilizing factors, possibly through direct interaction with the *DKK1* mRNA.

## RBM47 inhibits tumor progression by suppressing Wnt activity

DKK1 is an inhibitor of Wnt signaling, a pathway with a well-established role in regulating stem cell characteristics in both normal and malignant cells (*Clevers and Nusse, 2012*). Indeed, DKK1 as well as other Wnt antagonists have been shown to inhibit breast cancer progression (*Bafico et al., 2004*; *Cowling et al., 2007*; *Mikheev et al., 2008*; *Matsuda et al., 2009*). The fact that RBM47 was able to increase DKK1 secretion therefore suggested that RBM47 may also inhibit Wnt signaling and consequently reduce the tumorigenic fitness of metastatic breast cancer cells. We first tested the effects of RBM47 on cancer cell Wnt responsiveness by treating WT6 cells with recombinant Wnt3A and subsequently measuring the expression of *AXIN2*, a common TCF/β-catenin target gene and a general marker of Wnt activity (*Lustig et al., 2002*; *Clevers and Nusse, 2012*). Doxycycline-induced expression of RBM47 in WT6 cells led to a dampened Wnt3A-dependent *AXIN2* induction (*Figure 8A*). This inhibition of *AXIN2* expression was at least partially dependent on DKK1 (*Figure 8B*). In agreement with these results, human breast cancers with low RBM47 expression had in general higher levels of Wnt transcriptomic activity when compared to tumors with high RBM47 expression in the TCGA cohort (*Figure 8C*).

We then tested the possibility that RBM47 would suppress tumorigenesis by implanting low numbers of WT6 cells at the orthotopic site. The induction of RBM47 by doxycycline was able to significantly delay the emergence of mammary tumors (*Figure 8D*), and the tumors that formed were smaller in size (*Figure 8E*). This observation was confirmed in WT10 cells (*Figure 8—figure supplement 1A*). In line with this, inhibition of DKK1 expression in the MDA231-BrM2 cells promoted tumor formation in the orthotopic site (*Figure 8F*, *Figure 8—figure supplement 1B*). Finally, we used the two exon 1 DKK1 shRNAmiR constructs (*Figure 7C*) to reduce DKK1 levels in the WT6 cells (*Figure 8—figure supplement 1C*) and tested how this affected the brain metastatic ability of these cells in the presence of RBM47. Only 1/9 mice (16%) injected with control cells developed brain metastasis, whereas 8/17 mice (47%) inoculated with DKK1 RNAi construct transduced cells developed metastasis (*Figure 8G,H*). Taken together, these observations suggested that RBM47-dependent suppression of tumor progression was partially mediated by its ability to increase the production of the Wnt antagonist DKK1, a secreted protein that can inhibit tumor phenotypes in metastatic cancer cells.

## Discussion

Cancer genomes contain numerous genes with low-frequency mutations of unknown functional significance. We have studied one such gene, the previously uncharacterized *RBM47*, and demonstrate that it has tumor suppressive functions in breast cancer. RBM47 acts as a multifunctional RBP modulating alternative splicing and the abundance of several mRNAs, which can lead to inhibition of cancer progression (*Figure 8I,J*). These results highlight the significance of infrequent mutations in cancer, the importance of integrated experimental approaches to identify such functionally relevant mutations, and the role of broadly targeted mRNA chaperones as determinants of cancer progression.

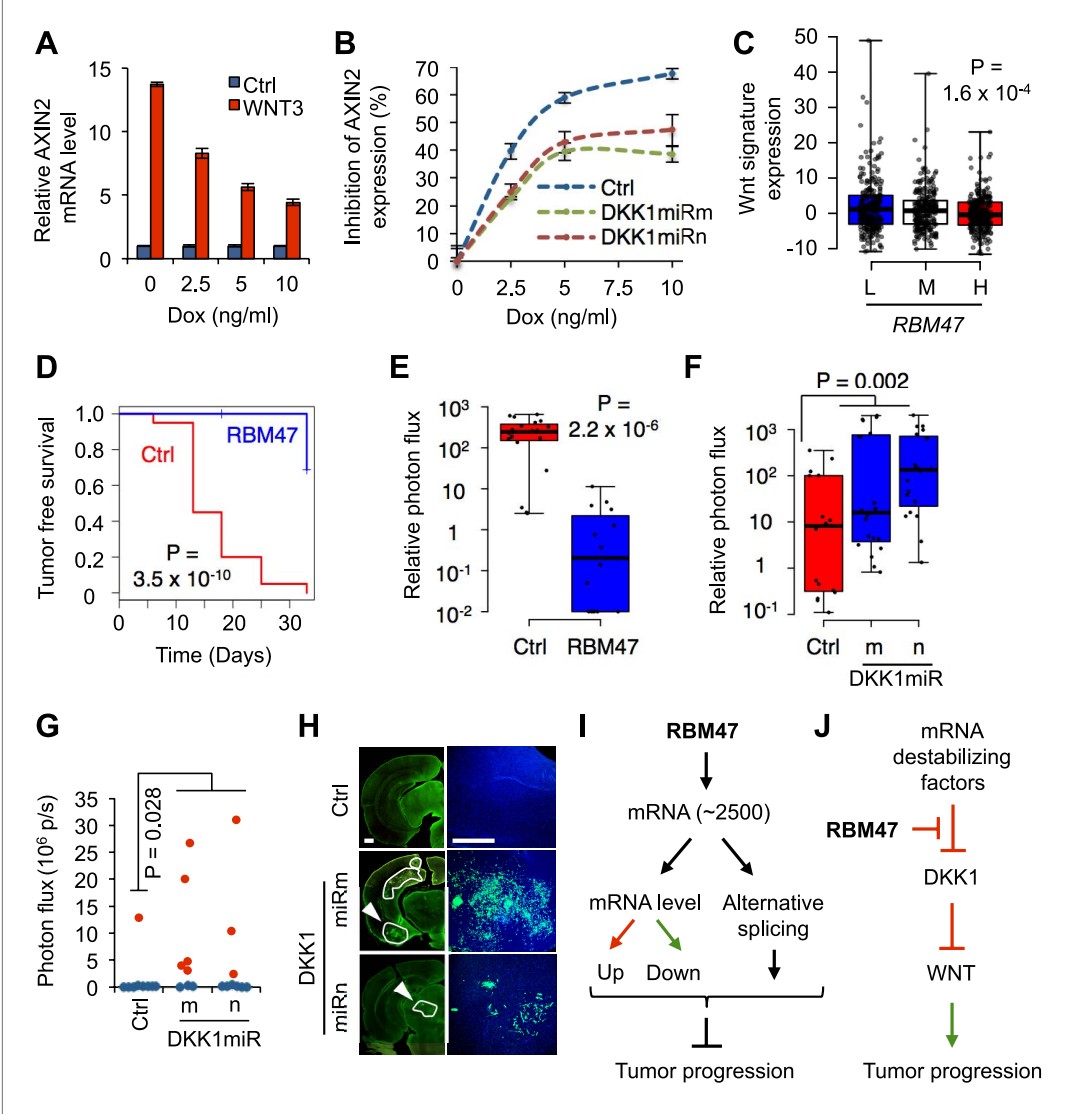

**Figure 8**. RBM47 suppresses tumor progression via Wnt inhibition. (**A**) *AXIN2* mRNA levels determined by quantitative real-time RT-PCR in WT6 cells treated with recombinant WNT3A in the presence of increasing concentrations of doxycycline. Error bars represent 95% confidence intervals obtained from multiple PCR reactions. (**B**) Normalized level of *AXIN2* mRNA inhibition as determined by quantitative real-time RT-PCR in WT6 cells transduced wither with control vector or shRNAmiR constructs targeting the first exon of *DKK1*. The cells were treated with recombinant WNT3A in the presence of increasing concentrations of doxycycline. Error bars represent 95% confidence intervals obtained from multiple PCR reactions. (**C**) Wnt pathway activity assessed in the TCGA cohort of primary breast tumors, grouped by *RBM47* expression tertiles (L, low; M, medium; H, high). Wnt signature value calculated as sum of z-scores for a curated set of 16 Wnt target genes in breast cancer. p-value determined by linear regression analysis. (**D**) Mammary tumor re-initiation assay. 5,000 WT6 cells implanted orthotopically in mice. RBM47 induced by doxycycline feed. Tumor growth detected by bioluminescence. p-value determined by the log-rank test. N = 20 tumors for each group. (**E**) Quantification of mammary tumor burden by in vivo bioluminescence imaging on day 33 of the experiment shown in (**D**). Data normalized to day 0 for each tumor. p-value calculated by the Wilcoxon rank-sum test. (**F**) Quantification of mammary tumor burden by in vivo bioluminescence imaging in mice inoculated with 231-Brm2 cells transduced with either control (pGIPZ) or DKK1-targeting shRNAmiR constructs. Data normalized to day 0 for each tumor. p-value calculated by one-tailed Wilcoxon rank-sum test. (**G**) Quantification of *ex vivo* brain bioluminescence shown for mice inoculated intracardiacly with WT6 cells transduced with either control (pGIPZ) or DKK1-targeting shRNAmiR constructs in the presence of RBM47, that is doxycycline in diet. One out of 9 (11%) control mice developed robust brain metastasis whereas 8/17 (47%) mice in the DKK1 knockdown groups showed metastasis. p-value calculated by one-tailed Student's *t* test. (**H**) Representative

*Figure 8. Continued on next page*

*Figure 8. Continued*

images of coronal brain sections analyzed for GFP immunofluorescence from the experiment shown in panel (**G**). Lesion contours are marked in white. Arrowheads indicate the lesions shown in higher magnification on the right; a similar brain area is shown for the control group. Scale bar 500 μm. (**I**) At the global level, RBM47 binds to ~2500 target mRNAs. However, the abundance or alternative splicing of only a fraction of these change depending on RBM47 status. The target genes represent molecules from various signaling pathways. The net effect of growth promoting and inhibiting alterations determine whether RBM47 loss is beneficial for a particular cancer clone. (**J**) At the target mRNA level, the effects of RBM47 are dependent on the presence of other factors that modulate mRNA processing. Hence, the phenotype of RBM47 loss depends on the intracellular molecular milieu on a per transcript basis. This is exemplified by the interaction of RBM47 with *DKK1* mRNA.

The following figure supplements are available for figure 8:

**Figure supplement 1**. DKK1 as a mediator of RBM47-dependent tumor suppression.

RBM47 contains three classical RNA recognition motifs (RRM domains). The closest homologs of RBM47 are Apobec1 complementation factor (A1CF) and hnRNP-Q, which regulate RNA editing (*Mehta et al., 2000*) and splicing and transcript stability, respectively (*Chen et al., 2008*; *Weidensdorfer et al., 2009*). In general, RBPs bind to and influence the function and fate of both pre-mRNAs and mRNAs (*Dreyfuss et al., 2002*). They can operate in large multiprotein complexes that dynamically regulate all the steps of mRNA biogenesis, nuclear export, stability and translation. Individual subunits of these complexes can therefore have diverse phenotypic roles depending on the exact protein complex they are in *Chaudhury et al. (2010)*. Our observations on RBM47 are in line with these known general principles of RBP function.

Our data demonstrate widespread and reproducible RBM47 binding to target mRNAs predominantly in introns and 3′UTRs, with the most robust binding occurring in 3′UTRs. Recent RNA-compete studies have proposed a binding motif for RBM47 in vitro (*Ray et al., 2013*). Our in vivo HITS-CLIP data does not suggest a clear nucleotide binding specificity for exogenous Flag-tagged RBM47, although some preference was observed for polyU stretches around CIMS sites. It has been shown that the presence of a canonical motif is neither necessary nor sufficient to predict HITS-CLIP binding sites of FUS in both mouse and human brain (*Lagier-Tourenne et al., 2012*), while specific sites suggested by in vitro RNA selection experiments are not enriched in HITS-CLIP derived FMRP binding sites in vivo (*Darnell et al., 2011*). This would suggest that other factors such as RNA accessibility, secondary structure or protein–protein interactions may modulate RBM47 target choice (*Li et al., 2014*). Further work is therefore needed for a comprehensive understanding of the determinants of RBM47-mRNA interactions.

We find that RBM47 binds robustly to ~2500 gene transcripts in human breast cancer cells, with only a subset showing steady state level change or alternative splicing upon RBM47 reintroduction. Given the stringent criteria used to define RBM47-bound and regulated targets and the generally low level of intronic RNA in a cell, it is likely that this subset is an underestimation of the number of RBM47-regulated transcripts, and does not take into account the potential for regulatory events, such as re-localization, that may not alter steady state transcript levels. The complex interplay of RBPs in agonistic and antagonistic modulation of mRNA is becoming increasingly apparent. For example, the RRM-domain containing protein HuR modulates the destabilizing effects of miRNAs (*Bhattacharyya et al., 2006*; *Kim et al., 2009*; *Young et al., 2012*) and AUF1 (*Zou et al., 2010*) on common target transcripts. The data presented here suggest that target-specific RBM47 regulation may arise through modulation of accessibility of other factors to a common mRNA transcript.

RBM47 binds and regulates transcripts that encode for proteins of several different biological functions. The effects of reduced RBM47 activity on cancer cell fitness, determined by the sum phenotypic output of all regulated target transcripts, may therefore vary depending on the context. Such pleiotropic effects could target multiple steps of cancer progression. Indeed, even though RBM47 loss was associated with metastatic cancer clones in our model systems, evidence for selection against RBM47 was detected already in primary breast cancers. One of the most highly bound RBM47 mRNA targets, the secreted Wnt inhibitor *DKK1*, is stabilized by RBM47 and partially mediates RBM47 tumor suppressive function. Interestingly, *rbm47* knockdown in zebra fish embryos leads to a headless phenotype mediated via up-regulation of the *wnt8a* pathway (*Guan et al., 2013*). In addition to DKK1, our analysis identified a number of potential mediators of RBM47 effects for future studies.

From a general perspective, the present findings illuminate two concepts. First, we show that low-frequency cancer mutations can give rise to tumorigenic phenotypes. Our work highlights the power of orthogonal approaches for the analysis of cancer genome resequencing data. Second, we show that loss of a broadly targeted and multifunctional RBP can increase the fitness of certain cancer cell clones in support of metastasis. This complements previous findings of RNA-binding proteins as mediators of oncogenic phenotypes (*Karni et al., 2007*; *Richard et al., 2008*; *Sommer et al., 2011*; *Das et al., 2012*; *Wang et al., 2013*). Deregulation of RNA-binding proteins is thus emerging as a prominent source of complex transcriptomic diversity that can serve as a platform for the selection of metastatic traits during tumor progression.

## Materials and methods

### Cell lines

The metastatic breast cancer cell lines have been previously described (*Kang et al., 2003*; *Minn et al., 2005*; *Bos et al., 2009*). SKBR3, ZR-75-30 and HCC1954 cells were obtained from ATCC (Manassas, VA). For retrovirus and lentivirus production, GPG29 and 293T cells, respectively, were utilized. All cell lines were maintained under standard tissue culture conditions. Single cell-derived clones were isolated utilizing fluorescence-activated cell sorting from genetically engineered 231-BrM2 (WT10, MUT3) and CN34-BrM2 (WT6) cells.

### cDNA expression and RNAi

For RBM47 restoration, *RBM47* was cloned into the pBABE-puro retroviral expression vector. The *RBM47$^{I281fs}$* was generated by site-directed mutagenesis. Virus was generated in the GPG29 packaging cells. For the generation of doxycycline-inducible expression constructs, both wild type and mutant FLAG-RBM47 were cloned into the pRetroX-Tight-Pur expression system (Clontech, Mountain View, CA).

For RNAi-mediated gene silencing, RBM47 pGIPZ shRNA constructs (clones V2LHS_176331 and V3LHS_393928, respectively) were obtained from Open Biosystems (Lafayette, CO). The DKK1 shRNAmiR constructs were designed based on the rules described by *Fellmann et al. (2011)* and cloned into the pGIPZ vector as described (*Dow et al., 2012*) using the following oligonucleotide templates:

DKK1miRm: TGCTGTTGACAGTGAGCGACGGGTCTTTGTCGCGATGGTATAGTGAAGCCACAGA TGTATACCATCGCGACAAAGACCCGGTGCCTACTGCCTCGGA
DKK1miRn: TGCTGTTGACAGTGAGCGACACCTTGAACTCGGTTCTCAATAGTGAAGCCACAGAT GTATTGAGAACCGAGTTCAAGGTGGTGCCTACTGCCTCGGA
DKK1miRo:TGCTGTTGACAGTGAGCGCCAACTCAATCCTAAGGATATATAGTGAAGCCACAGATG TATATATCCTTAGGATTGAGTTGATGCCTACTGCCTCGGA
DKK1miRp:TGCTGTTGACAGTGAGCGACAGTAAATTACTGTATTGTAATAGTGAAGCCACAGATG TATTACAATACAGTAATTTACTGCTGCCTACTGCCTCGGA
DKK1miRq:TGCTGTTGACAGTGAGCGAAACGGAAGTGTGATATGTTTATAGTGAAGCCACAGAT GTATAAACATATCACACTTCCGTTCTGCCTACTGCCTCGGA

The pGIPZ empty vector was used as a control.

### Animal studies

All animal experiments were performed in accordance with a protocol approved by MSKCC Institutional Animal Care and Use Committee. Lung metastasis assays were conducted in 5–7 week old female NOD/SCID mice. Brain metastasis and mammary tumor assays were carried out using 5–7 week old female athymic nude mice. In vivo bioluminescence imaging was performed using the IVIS Spectrum Xenogen machine (Caliper Life Sciences, Hopkinton, MA). For orthotopic mammary tumor assays, cells were mixed with Matrigel. Injections were confirmed and tumor growth was followed by bioluminescent imaging. Statistical significance of tumor and metastasis free survival was assessed by the log-rank test. Differences in raw and normalized bioluminescence signal was assessed by the Student's *t* test and Wilcoxon rank-sum test, respectively. Brain images were acquired with a Leica SP5 up-right confocal microscope and Zeiss AxioVert 200 M using 20X and 5X objectives. Image analysis was performed with Metamorph and ImageJ softwares.

### mRNA and protein detection

Total RNA was extracted using PrepEase RNA spin kit (USB, Cleveland, OH). We used Transcriptor First Strand cDNA Synthesis Kit (Roche, Indianapolis, IN) for cDNA synthesis. Quantitative PCR was

performed using predesigned Taqman gene expression assays (Life Technologies, Carlsbad, CA) and the 7900HT or ViiA 7 real-time PCR systems (Applied Biosystems/Life Technologies). *TBP* was used as a housekeeping control gene. mRNA stability was assessed by performing quantitative PCR after actinomycin D treatment. For immunoblotting, antibodies recognizing RBM47 (HPA006347; Sigma, St. Luis, MO), α-tubulin (11H10; Cell Signaling, Danvers, MA) and ACTB (Sigma) were utilized. Secondary antibodies were HRP (Pierce, Rockford, IL) or fluorescence (LiCor, Lincoln, NE) conjugated. Immunostaining for RBM47 (HPA006347; Sigma) was performed according to standard protocols in the MSKCC Molecular Cytology Core Facility on paraffin embedded tissue blocks. Secreted protein was detected by ELISA (R&D, Minneapolis, MN).

## HITS-CLIP

231BrM2 tet-on FLAG-RBM47 cells were treated with 1 ng/ml doxycycline for 3 days before 254 nm UV crosslinking at 400 mJ/cm$^2$ on a bed of ice (Stratalinker2400; Stratagene, La Jolla, CA). Samples were processed for HITS-CLIP as previously described (*Licatalosi et al., 2008*) using an anti-Flag anti-body (F3165; Sigma), and omitting 3' linker ligation in favor of direct labeling of protein-bound RNA with $^{32}$P-γ-ATP. Non-crosslinked cells were used as a negative control, with IPed Flag-RBM47 protein detected using a second anti-Flag antibody (F7425; Sigma). Purified RBM47-bound RNA fragments were polyA tailed (E-PAP; NEB), and reverse transcribed (Superscript III; Invitrogen, Carlsbad, CA) in the presence of Br-dUTP (Sigma). Unique polydT-NV RT-primers were used per replicate, containing Solexa sequences separated by an abasic furan (that serves as an ApeI cut site), a 6 nt degenerate region and a 6 nt index sequence to allow for multiplexing during sequencing.

> RT Primer 1pGCACTGTTN$_6$GATCGTCGGACTGTAGAACTCT/idSp/CAAGCAGAAGACGGCATACGAT$_{20}$VN
> RT Primer 2pGCGAAACTN$_6$GATCGTCGGACTGTAGAACTCT/idSp/CAAGCAGAAGACGGCATACGAT$_{20}$VN

BrdU-cDNA was stringently purified by IP (sc-32323; Santa Cruz, Dallas, TX) using ProteinG dyna-beads (Invitrogen), eluted from the beads via BrdU competitive elution (Sigma), and re-immunoprecip-itated. cDNA was circularized on bead (CircLigase ssDNA Ligase II, Epicentre, Madison, WI), washed and digested with ApeI (NEB, Ipswich, MA) to relinearize. cDNAs were eluted from the beads by heating to 98C in Phusion HF Buffer (NEB), then PCR amplified using Phusion DNA polymerase (NEB) and SYBR Green I (Invitrogen) in an iQ5 real-time PCR machine in order to monitor amplification, with the samples being removed when the RFU signal reached ~1000.

> P5—aatgatacggcgaccaccgacaggttcagagttctacagtccgacg
> P3—caagcagaagacggcata

PCR products were purified using MinElute columns (Qiagen, Valencia, CA) as per manufacturer's instructions and quantified using Quant-It (Invitrogen). cDNA was multiplexed and sequenced using Illumina Hi-Seq (small RNA sequencing primer—cgacaggttcagagttctacagtccgacgatc). All data analysis was done using the Galaxy platform (*Hillman-Jackson et al., 2012*), as previously described (*Licatalosi et al., 2008*; *Chi et al., 2009*; *Darnell et al., 2011*; *Zhang and Darnell, 2011*).

## RT-PCR validation

RT-PCR validiation was carried out using total RNA from MDA231-BrM2 (Control) and WT10 cells (iScript, Bio-Rad, Accuprime Pfx Supermix 1, Life Technologies) as previously described (*Licatalosi et al., 2012*). In all cases lanes 1–3 contain an equal mixture of control and WT10 cell cDNA amplified at *n-1*, *n* and *n+1* cycles, lane 4 contains an absence of reverse transcriptase control (−RT), with all other lanes corresponding to replicate samples of the indicated cell type amplified to *n* cycles. IR and ΔI calculated using ImageJ (*Schneider et al., 2012*).

## Bioinformatic analysis

All analyses were conducted using R. Microarray data from human untreated tumor data sets (GSE2603 [*Minn et al., 2005*] and GSE2034 [*Wang et al., 2005*]) were preprocessed as described (*Zhang et al., 2009*). For the RNA-seq TCGA data set, normalized mRNA z-scores were downloaded from the TCGA cBio portal (*Cerami et al., 2012*). The microarray data from metastatic cell lines (*Minn et al., 2005*; *Bos et al., 2009*) were processed with GCRMA together with updated probe set definitions using R packages *affy*, *gcrma* and *hs133ahsentrezgcdf* (version 10). Unsupervised hierar-chical clustering was performed using the function *heatmap.2* with Pearson's correlation coefficient

as the similarity metric. For survival analysis, a Cox proportional hazards model was utilized as implemented in the *coxph* function in the R-package *survival*. For RNA-seq analysis of the cell lines, raw paired-end sequencing data were mapped to human genome (hg19 build) with STAR2.3.0 (*Dobin et al., 2013*) using standard options. Reads mapped to each transcript were counted by HTSeq v0.5.4 (*Anders and Huber, 2010*) with default settings. The read count table was normalized to library size by DESeq (*Anders and Huber, 2010*). Correlation with RBM47 expression was assessed by Pearson's correlation coefficient utilizing the R functions *cor* and *cor.test*. Wnt pathway activity in clinical tumors was assessed utilizing a curated list of Wnt target genes in breast cancer (*Matsuda et al., 2009*) and calculating sums of z-scores for each tumor.

## Accession numbers
RNA-seq and HITS-CLIP data have been deposited to the Gene Expression Omnibus under the accession numbers GSE53779 and GSE58381.

## Acknowledgements

We thank members of the Massagué lab for discussion. This work was supported by a grant from the Starr Cancer Consortium to RBD and JM, JM and RBD are investigators of the Howard Hughes Medical Institute.

## Additional information

### Competing interests
RBD: Reviewing editor, *eLife*. JM: Reviewing editor, *eLife*. The other authors declare that no competing interests exist.

### Funding

| Funder | Grant reference number | Author |
|---|---|---|
| Howard Hughes Medical Institute (HHMI) | | Joan Massagué |
| Starr Cancer Research Foundation | SCC I4-A435 | Joan Massagué |
| K Albin Johanssons Stiftelse | | Sakari Vanharanta |

The funder had no role in study design, data collection and interpretation, or the decision to submit the work for publication.

### Author contributions
SV, CBM, Conception and design, Acquisition of data, Analysis and interpretation of data, Drafting or revising the article; WS, MV, AM, Acquisition of data; YZ, Acquisition of data, Analysis and interpretation of data; RBD, JM, Conception and design, Analysis and interpretation of data, Drafting or revising the article

### Ethics
Animal experimentation: This study was performed in strict accordance with the recommendations in the Guide for the Care and Use of Laboratory Animals of the National Institutes of Health. All of the animals were handled according to approved institutional animal care and use committee (IACUC) protocols (#99-09-032) of Memorial Sloan Kettering Cancer Center. Tumor cell inoculation and imaging was performed under ketamine/xylazine anesthesia, and every effort was made to minimize suffering.

## Additional files

### Supplementary files
• Supplementary file 1. RBM47-bound alternatively spliced exons showing RBM47-dependent ΔI ≥|0.2|.

• Supplementary file 2. mRNAs changed upon RBM47 reintroduction.

• Supplementary file 3. RBM47 bound transcripts.

## Major datasets

The following datasets were generated:

| Author(s) | Year | Dataset title | Dataset ID and/or URL | Database, license, and accessibility information |
|---|---|---|---|---|
| Vanharanta S, Yilong Zou, Massagué J | 2014 | The RNA-binding protein RBM47 suppresses metastatic breast cancer progression | GSE53779; http://www.ncbi.nlm.nih.gov/geo/query/acc.cgi?acc=GSE53779 | Publicly available at the Gene Expression Omnibus (http://www.ncbi.nlm.nih.gov/geo/). |
| Marney CB, Darnell RB | 2014 | The identification of RBM47 binding sites and RBM47-dependent alternative splicing events in brain metastatic breast cancer cells | GSE58381; http://www.ncbi.nlm.nih.gov/geo/query/acc.cgi?acc=GSE58381 | Publicly available at the Gene Expression Omnibus (http://www.ncbi.nlm.nih.gov/geo/). |

The following previously published datasets were used:

| Author(s) | Year | Dataset title | Dataset ID and/or URL | Database, license, and accessibility information |
|---|---|---|---|---|
| Bos PD, Massagué J | 2009 | Genes that mediate breast cancer metastasis to the brain | GSE12237; http://www.ncbi.nlm.nih.gov/geo/query/acc.cgi?acc=GSE12237 | Publicly available at the Gene Expression Omnibus (http://www.ncbi.nlm.nih.gov/geo/). |
| Minn AJ, Massagué J | 2005 | Subpopulations of MDA-MB-231 and Primary Breast Cancers | GSE2603; http://www.ncbi.nlm.nih.gov/geo/query/acc.cgi?acc=GSE2603 | Publicly available at the Gene Expression Omnibus (http://www.ncbi.nlm.nih.gov/geo/). |
| Wang Y, et al. | 2005 | Breast cancer relapse free survival | GSE2034; http://www.ncbi.nlm.nih.gov/geo/query/acc.cgi?acc=GSE2034 | Publicly available at the Gene Expression Omnibus (http://www.ncbi.nlm.nih.gov/geo/). |
| The Cancer Genome Atlas Network | 2012 | Comprehensive molecular portraits of human breast tumours | http://cbioportal.org | cBio Cancer Genomics Portal. |

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
