## [Decision Letter]

Thank you for sending your work entitled “Loss of the multifunctional RNA-binding protein RBM47 as a source of selectable metastatic traits in breast cancer” for consideration at *eLife*. Your article has been favorably evaluated by Tony Hunter (Senior editor) and 2 reviewers, one of whom is a member of our Board of Reviewing Editors.

The Reviewing editor and the other reviewers discussed their comments before we reached this decision, and the Reviewing editor has assembled the following comments to help you prepare a revised submission.

The reviewers felt that the identification of the RBM47 RNA-binding protein as a regulator of metastasis was an important contribution that could be published in *eLife*. However, the following issues would need to be addressed before we can consider the work further for publication in *eLife*.

Major comments:

1) A critical issue is whether RBM47 has a specific role in regulating metastasis as opposed to more broadly affecting cell proliferation and growth. More specifically, Figures 2 and 3 attempt to show that RBM47 suppresses experimental metastases in the brain and the lung. However, without examining whether RBM47 overexpression and knockdown also affect cell proliferation in culture and primary tumor growth in mice, it is unclear whether RMB47 specifically regulates metastasis vs. cell proliferation. Indeed, Figure 8 shows that expression of RBM47 drastically inhibits primary tumor growth, suggesting a possible role of RBM47 in regulating primary tumor growth, with a possible downstream effect on metastasis. Figure 8 aims to demonstrate a critical role of DKK1 downstream of RBM47 in suppressing tumor progression. Since Figure 8 shows that DKK1 knockdown resulted in significantly bigger primary tumors, it is again unclear whether there is any specific effect on metastasis. For this work to be of sufficient impact for publication in *eLife* experiments to demonstrate a specific role of RBM47 in metastasis, as opposed to general cell proliferation, are required.

2) Given the pleiotropic effects of the RBM47 protein on cellular RNA metabolism, it is possible that multiple target RNAs are involved and stronger evidence that the DKK1 and Wnt signaling pathways are the primary targets would be required. Possible experiments include:

a) Examining whether RBM47 expression is inversely correlated with Wnt gene signatures in the human breast tumor datasets described in Figure 1.

b) Examining if weaker DKK1 shRNAs have the appropriate effect on tumor growth and metastasis as related to RBM47. This is based on the observation that Figure 8—figure supplement 1 shows that RBM47 overexpression resulted in ∼3-fold increase of DKK1 protein level. However, DKK1 shRNAs used in suppression of DKK1 protein to an almost undetectable level; much lower than the level in the control cells without RBM47. Therefore, the observed tumor growth and brain metastasis formation upon DKK1 knockdown (Figure 8) may not be necessarily related to RBM47. To test this, much weaker DKK1 shRNAs should be used to bring down the DKK1 protein level to the level in the control cells as shown in Figure 8 to test its effect on tumor growth and metastasis. Ideally, depletion of the relevant Wnts expressed by these cells would establish the importance of DKK1 as a target for metastasis suppression by RBM47.

3) Most of the experiments were carried out with the metastatic variants of the MDA-MB-231 cell line, and one would like to know how general the effect of manipulating RBM47 levels on breast cancer cell metastasis is. Some experiments were carried out with CN34 cells, a metastatic breast cancer line derived from a pleural effusion at MSKCC, and these data support the conclusion that RBM47 acts to reduce metastasis. In this regard, the authors showed that CN34-BrM2 cells have low levels of RBM47 – did they check if RBM47 expression reduces the level of brain metastases? Have any other breast cancer cell lines been assessed for the effects of RBM47 expression or depletion on metastatic potential (e.g. lines like SKBR3 that have high endogenous RBM47)?

Minor comments:

4) It would improve the CLIP data to normalize their number of tags to the abundance of that RNA from the RNA-seq data. This might reveal a more relevant binding to intron sequences (which are generally present at low levels).

5) It is possible that effects on alternative splicing can influence RNA stability due to sequence differences in mature mRNAs. Although not required for publication, it would improve the manuscript if the authors determined whether there is a significant overlap between mRNAs with altered splicing and abundance in order to see if these two sets are related.

6) The CLIP experiment was done with over-expressed RBM47 protein and this has the potential to influence the observed binding sites. This caveat should at least be made explicit in the manuscript.

7) The key functional data presented in this study (Figures 2, 3 and 8) relied on overexpressing RBM47 in tumor cells. It is unclear how the level of RBM47 overexpression is compared to the physiological level of RBM47 in normal cells or the parental non-metastatic cells. This information is critical to determine whether the observed tumor growth reduction upon RBM47 overexpression is due to physiological effect of RBM47 or due to toxic effects of overexpressed RBM47.

8) It would significantly strengthen the manuscript if the effect of RBM47 on metastasis could be ascribed to a specific step in metastasis (e.g. invasion, survival, stem cell properties, interaction with stromal cells, or per-metastatic niche).

9) Since no biochemical analysis was performed on the RBM47 I281fs mutant, it is unclear whether this mutation completely abolishes RBM47 function. Therefore, in Figure 2, LM2 cells expressing a vector control should also have been used to compare the effect on expressing WT and MUT RBM47.

10) In Figure 2 shRNAs were used to knock down RBM47 in LM2 cells to determine the effects on lung colonization. Western blotting analysis should be presented to show the RBM47 knockdown levels. Instead, it is confusing why Figure 3 shows the western blot with CN34 cells.

11) Figure 3 should include staining for RBM47 in the parental cells or cells expressing a control vector. Otherwise, it is unclear what is the negative RBM47 staining. Figure 3 states these are RBM47 negative tumors, although many show positive signals. It is important to co-stain for a marker specific for tumor cells, so that expression of RBM47 in tumor cells vs. stromal cells can be separated.

---

## [Author Response]

We would like to thank the reviewers for their time and insightful comments. We were happy to see that all of you found our work interesting and that the general assessment was positive. The comments highlighted important points, all of which we have now addressed. We feel that this has improved our paper. We have also decided to eliminate the term hnRNP from the paper, as it is an outdated concept, originally meant to refer to abundant nuclear RNA binding proteins that coat hnRNA. However, the term essentially has no current meaning except for historical purposes.

*1) A critical issue is whether RBM47 has a specific role in regulating metastasis as opposed to more broadly affecting cell proliferation and growth. More specifically,*
Figures 2 and 3
*attempt to show that RBM47 suppresses experimental metastases in the brain and the lung. However, without examining whether RBM47 overexpression and knockdown also affect cell proliferation in culture and primary tumor growth in mice, it is unclear whether RMB47 specifically regulates metastasis vs. cell proliferation. Indeed,*
Figure 8
*shows that expression of RBM47 drastically inhibits primary tumor growth, suggesting a possible role of RBM47 in regulating primary tumor growth, with a possible downstream effect on metastasis.*
Figure 8
*aims to demonstrate a critical role of DKK1 downstream of RBM47 in suppressing tumor progression. Since*
Figure 8
*shows that DKK1 knockdown resulted in significantly bigger primary tumors, it is again unclear whether there is any specific effect on metastasis. For this work to be of sufficient impact for publication in eLife experiments to demonstrate a specific role of RBM47 in metastasis, as opposed to general cell proliferation, are required*.

We agree with the assessment that the data presented in the manuscript does not show specific effects of RBM47 as a suppressor of metastatic progression. In fact, the data in Figure 8 clearly show effects on mammary tumor formation as well. This is in line with the data from clinical patient cohorts: RBM47 mutations have been identified in primary breast cancers. Thus, we do not believe that RBM47 inactivation would result in a specific pro-metastatic phenotype. However, as described if Figure 1 and associated text, we did identify RBM47 through an analysis of metastatic breast cancer model systems and clinical datasets. This showed that RBM47 expression is reduced in highly metastatic subpopulations of breast cancer cells when compared to their weakly metastatic but highly tumorigenic parental lines, and that low *RBM47* expression correlates with poor patient outcome. Moreover, in the breast cancer brain metastasis sequenced by Ding et al. (2010), the RBM47 mutation was highly enriched when compared to the primary tumor. Combined with our functional data, all this evidence suggests that the most aggressive cancer cell populations that are most likely to form metastases benefit from RBM47 loss and hence RBM47 loss is further selected for during metastatic progression.

We would like to address here the widespread idea that ‘true’ metastasis genes and functions are not involved at all in primary tumor formation. In fact, with the exception of genes that mediate the last stage of the metastatic process – overt colonization of specific organ microenvironments – the majority of genes implicated in metastasis confer advantages to cancer cells already in a primary tumor. Otherwise, clones overexpressing such pro-metastatic genes would not become selected and enriched enough in primary tumors for their expressed genes to score as predictive metastasis markers or signatures. These genes are not tumor initiation or early progression oncogenes. Their functions provide a selective advantage to invasive cancer cells that venture into hostile microenvironments, for example, at the invasive front. The clones expressing these genes are thereby primed for metastatic seeding in distant sites, where their advantageous traits may undergo additional selection and amplification. Extensive clinical and experimental evidence for this concept has been presented in our work (1-5) and the work of others. Therefore, the idea that authentic pro-metastatic genes contribute to tumorigenesis only in distant organs and not at all in primary tumors is incorrect, in our view. We have succinctly annotated the revised manuscript in order to clarify this issue.

The evolutionary process that shapes tumor progression at the primary site pre-selects phenotypes that will later on enhance the metastatic fitness of the disseminated cells. As the vast majority of disseminated cancer cells die, this pre-selection is essential for cancer cell survival at the early stages of metastasis initiation. For this reason, the majority of important mediators of metastasis identified to date also have detectable effects on tumor growth at the primary site. This is the case with RBM47 as well.

The most important finding of our paper is the demonstration that loss of RBM47, a previously uncharacterized RNA chaperone with widespread binding across the transcriptome, is selected for during metastatic cancer progression. We show that this is partially mediated by enhanced activation the Wnt signaling pathway, but we believe that other pro-tumorigenic signaling pathways could also be involved in the process. Our intention therefore is not to suggest that RBM47 has specific functions suppressing metastasis, but rather that its loss contributes to the establishment of the most aggressive cancer phenotypes that are vital to metastatic dissemination. This is an important concept that highlights the importance of alterations in transcriptomic homeostasis as a source of phenotypic diversity required for cancer progression. In order to clarify this, we have reworded the Abstract and manuscript.

*2) Given the pleiotropic effects of the RBM47 protein on cellular RNA metabolism*, *it is possible that multiple target RNAs are involved and stronger evidence that the DKK1 and Wnt signaling pathways are the primary targets would be required. Possible experiments include:*

*a) Examining whether RBM47 expression is inversely correlated with Wnt gene signatures in the human breast tumor datasets described in*
Figure 1.

*b) Examining if weaker DKK1 shRNAs have the appropriate effect on tumor growth and metastasis as related to RBM47. This is based on the observation that*
Figure 8
*shows that RBM47 overexpression resulted in ∼3-fold increase of DKK1 protein level. However, DKK1 shRNAs used in suppression of DKK1 protein to an almost undetectable level; much lower than the level in the control cells without RBM47. Therefore, the observed tumor growth and brain metastasis formation upon DKK1 knockdown (*Figure 8*) may not be necessarily related to RBM47. To test this, much weaker DKK1 shRNAs should be used to bring down the DKK1 protein level to the level in the control cells as shown in*
Figure 8
*to test its effect on tumor growth and metastasis. Ideally, depletion of the relevant Wnts expressed by these cells would establish the importance of DKK1 as a target for metastasis suppression by RBM47*.

Our work indicates that RBM47 has multiple target molecules. This is in line with the known principles of nuclear RNA binding protein function; they act as part of large protein complexes to facilitate multiple aspects of mRNA biogenesis. Indeed, we provide evidence that RBM47 regulates both mRNA stability and splicing. As the various target mRNAs of RBM47 encode for proteins that function in several biological processes, the phenotypic output of RBM47 loss will therefore have pleiotropic effects. Our data in Figure 7 show the importance of other mRNA processing factors as regulators of RBM47 phenotypic output underscoring the fact that loss of RBM47 can have context-dependent effects. Even with the widespread binding pattern of RBM47, we demonstrate here that cells with reduced RBM47 function are repeatedly selected for during tumor progression. Thus, at least in some circumstances, the net effect of RBM47 loss is beneficial for a cancer cell.

In the current study, we focused on DKK1 as one of the downstream mediators of RBM47 action. DKK1 was chosen for two reasons: (i) it was one of the most prominent target transcripts of RBM47 as determined by HITS-CLIP and (ii) it is a known modulator of the Wnt pathway. We show that RBM47 can indeed inhibit Wnt signaling by stabilizing DKK1. We also demonstrate that DKK1 knockdown is able to partially rescue the tumor suppressive effects of RBM47. These results suggest, as pointed out by the referees, that RBM47 expression should correlate negatively with a Wnt responsive signature in clinical data sets. In agreement with the prediction, we have now found that tumors with low RBM47 expression tend to have higher expression of Wnt target genes, as determined by pathway analysis using a curated breast cancer Wnt signature (6). Following the option 2(a) suggested by the referees, these new data have been added into the new Figure 8 and associated text as requested.

Despite providing evidence of DKK1 as a mediator of the pro-tumorigenic effects of RBM47 loss, we do not mean to suggest that DKK1 is the only important target transcript of RBM47. DKK1 is but one of many RBM47 targets, the balance of which makes a *net* positive contribution to metastasis, much in the same way that dysfunctions in other epigenetic regulators do. In this paper we use DKK1 as an important example, but acknowledge the possible contributions of other RMB47 targets.

The upshot of our work is that the loss of a promiscuous RNA-binding protein can, despite its large number of functionally unrelated target mRNAs, lead to the emergence of pro-metastatic cancer cell traits. This finding complements the recent findings of mutations in epigenetic regulators in cancer, and highlights the importance of widespread perturbations in RNA homeostasis as a source of cancer cell phenotypes. In order to clarify this point, the revised manuscript highlights other potential mediators of RBM47 effects for further study..

*3) Most of the experiments were carried out with the metastatic variants of the MDA-MB-231 cell line, and one would like to know how general the effect of manipulating RBM47 levels on breast cancer cell metastasis is. Some experiments were carried out with CN34 cells, a metastatic breast cancer line derived from a pleural effusion at MSKCC, and these data support the conclusion that RBM47 acts to reduce metastasis. In this regard, the authors showed that CN34-BrM2 cells have low levels of RBM47* – *did they check if RBM47 expression reduces the level of brain metastases? Have any other breast cancer cell lines been assessed for the effects of RBM47 expression or depletion on metastatic potential (e.g*. *lines like SKBR3 that have high endogenous RBM47)?*

We identified RBM47 as a suppressor of breast cancer progression through an integrative approach that combined gene expression information from metastatic breast cancer models with mutational and gene expression data from clinical datasets. Importantly, the clinical cohorts utilized in this study comprise hundreds of tumor samples, providing critical evidence that the observations made in our experimental model systems have more general clinical relevance. To this end, the new analysis in Figure 8 further strengthens our conclusions by providing additional evidence of the negative correlation between RBM47 expression and Wnt pathway activity.

We agree that increasing the number of model systems increases confidence in the general relevance biological data. Our study therefore contains experimental results from several cell line models (MDA231, CN34, SKBR3 and ZR-75-30) as well as clonal metastatic derivatives, including one derived from the CN34-BrM2 cells (WT6). The results on the effects of RBM47 on the brain metastatic potential of these cells is shown in Figure 3. The cell lines from which the three single cell-derived brain metastatic clones used in this study were isolated has been clarified in the Methods section of the revised manuscript. In sum, we feel that the combined power of large clinical data sets and several experimental models give sufficient confidence in the generality of our findings.

Minor comments:

*4) It would improve the CLIP data to normalize their number of tags to the abundance of that RNA from the RNA-seq data. This might reveal a more relevant binding to intron sequences (which are generally present at low levels)*.

We agree that normalizing HITS-CLIP data to total RNA reads could reveal more prominent intronic binding patterns. However, our existing RNAseq data corresponding to the HITS-CLIP experiments shown in Figures 4 and 5 are polyA-selected and therefore, given the low level of reads mapping to introns, it is not suitable for intronic HITS-CLIP normalization. In order to take the low level of intronic RNA into account, and anticipating that pre-mRNA may be present at levels approximately 1000x lower than mRNA (7), we have lowered the threshold for determining robust RBM47 binding in introns while maintaining a conservative cut-off of total HITS-CLIP reads in clusters in the vicinity of alternatively spliced exons (see Figure 4). While this approach increases the sensitivity for detecting intronic binding, we do agree that our data may underestimate the degree to which RBM47 intronic binding is associated with RBM47-dependent alternative splicing. To clarify this, we have amended the text in the revised manuscript.

*5) It is possible that effects on alternative splicing can influence RNA stability due to sequence differences in mature mRNAs. Although not required for publication, it would improve the manuscript if the authors determined whether there is a significant overlap between mRNAs with altered splicing and abundance in order to see if these two sets are related*.

Alternative splicing can alter mRNA stability, e.g. through the introduction of cryptic NMD exons (8). However, at the cutoff levels used in this study, we did not see any overlap between the transcripts that underwent alternative splicing and genes that showed altered message levels upon RBM47 reintroduction. As discussed in Response 4, deeper intronic RNA sequencing could identify additional RBM47-dependent splicing events, which could also reveal links to mRNA stability, but based on current data, alternative splicing does not seem to be a major determinant of altered mRNA stability in the context of RBM47.

*6) The CLIP experiment was done with over-expressed RBM47 protein and this has the potential to influence the observed binding sites. This caveat should at least be made explicit in the manuscript*.

The text has been revised as requested.

*7) The key functional data presented in this study (*Figures 2, 3 and 8*) relied on overexpressing RBM47 in tumor cells. It is unclear how the level of RBM47 overexpression is compared to the physiological level of RBM47 in normal cells or the parental non-metastatic cells. This information is critical to determine whether the observed tumor growth reduction upon RBM47 overexpression is due to physiological effect of RBM47 or due to toxic effects of overexpressed RBM47*.

We agree that protein overexpression is a potential source of artifacts. In order to control for this, however, we have taken several measures:

a) shRNA constructs were used to validate the metastasis suppressive effects of RBM47 in vivo (Figure 2 and Figure 2—figure supplement 1), as well as the capability of RBM47 loss to regulate DKK1 mRNA levels (Figure 7 and Figure 7—figure supplement 1).

b) Single cell-derived metastatic cell lines with doxycycline-inducible RBM47 expression were generated. This allowed us to control RBM47 protein levels, as can be seen in Figure 3—figure supplement 2. The effects on DKK1 expression were detectable already at RBM47 protein levels lower or similar to that seen in control cells expressing only endogenous RBM47 (doxycycline 2.5ng/ul in WT6 cells, Figure 3—figure supplement 2 and Figure 6), even though no effect was seen in cell proliferation in vitro, as shown by the new piece of data in Figure 3—figure supplement 2.

c) In the RNA-seq analysis, stringent conditions were applied to ensure that the RBM47 responsive signature consisted of genes that were changed already at the lowest detectable level of RBM47 protein expression. This signature was able to classify human cancer specimens in a way that correlated with RBM47 expression, as is shown in Figure 6 and Figure 6—figure supplement 1.

In addition, DKK1 knockdown was able to partially rescue the effects of RBM47 reintroduction in the brain metastasis assay (Figure 8). Thus, we feel that the collective evidence, including data from both loss-of-function and gain-of-function genetic experiments as well as bioinformatic analysis of large clinical datasets, support the conclusions of the paper.

*8) It would significantly strengthen the manuscript if the effect of RBM47 on metastasis could be ascribed to a specific step in metastasis (e.g. invasion, survival, stem cell properties, interaction with stromal cells, or per-metastatic niche)*.

As discussed in Response 2, RBM47 like all nuclear RNA binding proteins is likely to function in multiprotein complexes to facilitate mRNA biogenesis. In line with this, we find that RBM47 binds to multiple target mRNAs. These targets do not belong to a distinct pathway or molecular class, but rather, represent transcripts that encode for proteins that have functional roles in several biological processes. We focused on DKK1 as one of the many target molecules of RBM47 and show that RBM47 can modulate Wnt signaling via DKK1 stabilization. The Wnt pathway can modulate multiple breast cancer phenotypes including proliferation (9-11), migration (6) and stem cell fitness (12). In line with this, we show in Figure 8 that RBM47 can both inhibit tumor re-initiation and growth. However, it remains possible that further phenotypic characterization could reveal additional functional consequences of RBM47 loss. Thus, given the pleiotropic nature of RBM47 function, we believe that ascribing the tumor suppressive functions of RBM47 to a single step in the metastatic cascade would be an over-simplification of its biological function.

*9) Since no biochemical analysis was performed on the RBM47 I281fs mutant, it is unclear whether this mutation completely abolishes RBM47 function. Therefore, in*
Figure 2*, LM2 cells expressing a vector control should also have been used to compare the effect on expressing WT and MUT RBM47*.

Even though I281fs mutant mRNA was expressed at similar levels than wildtype *RBM47* (see Figure 2—figure supplement 1), Western blot analyses indicate that the RBM47 I281fs mutant protein is not expressed at doxycycline dosages that induce wildtype RBM47 expression, as can be seen in the data provided in new Figure 2—figure supplement 1. Furthermore, the experiments shown in Figure 3 demonstrate that doxycycline-induced expression of I281fs does not have any effect on brain metastasis formation by the MUT3 cells. Collectively this suggests that the RBM47 I281fs mutant expressing cells function as a negative control.

*10) In*
Figure 2
*shRNAs were used to knock down RBM47 in LM2 cells to determine the effects on lung colonization. Western blotting analysis should be presented to show the RBM47 knockdown levels. Instead, it is confusing why*
Figure 3
*shows the western blot with CN34 cells*.

As requested, the Western blot related to Figure 2 has been added to Figure 2—figure supplement 1. For clarity, the Western blot for CN34 cells has been moved to Figure 2—figure supplement 1.

*11)*
Figure 3
*should include staining for RBM47 in the parental cells or cells expressing a control vector. Otherwise, it is unclear what is the negative RBM47 staining.*
Figure 3
*states these are RBM47 negative tumors, although many show positive signals. It is important to co-stain for a marker specific for tumor cells, so that expression of RBM47 in tumor cells vs. stromal cells can be separated*.

The I281fs mutation introduces a premature stop codon. For this reason, the epitope against which the RBM47 antibody was developed is missing from the mutant protein. RBM47 mutant expressing cells can therefore be used as a negative control, as is shown in Figure 3. We do agree, however, that using the word ‘negative’ in this context is inappropriate. The text has been revised to more accurately describe the staining as ‘weak’ or ‘low’. As requested, we have also complemented the analysis by including data on a tumor-specific marker. The revised Figure 3—figure supplement 1 and associated text now show examples of metastatic nodules that have either high or low RBM47 expression as determined by immunohistochemistry against RBM47 and human vimentin. We have also assessed tumor cell proliferation in these tumors using an antibody against Ki67. These data show that both the RBM47 high and low nodules proliferate at similar rates, further emphasizing the conclusion that while some metastases do reduce the transgene expression, others are capable of developing in the presence of RBM47.

References

1) Tavazoie, S.F., et al., *Endogenous human microRNAs that suppress breast cancer metastasis.* Nature, 2008. **451**(7175): p. 147-52.

2) Gupta, G.P., et al., *Mediators of vascular remodelling co-opted for sequential steps in lung metastasis.* Nature, 2007. **446**(7137): p. 765-70.

3) Bos, P.D., et al., *Genes that mediate breast cancer metastasis to the brain.* Nature, 2009. **459**(7249): p. 1005-9.

4) Minn, A.J., et al., *Genes that mediate breast cancer metastasis to lung.* Nature, 2005. **436**(7050): p. 518-24.

5) Vanharanta, S., et al., *Epigenetic expansion of VHL-HIF signal output drives multiorgan metastasis in renal cancer.* Nat Med, 2013. **19**(1): p. 50-6.

6) Matsuda, Y., et al., *WNT signaling enhances breast cancer cell motility and blockade of the WNT pathway by sFRP1 suppresses MDA-MB-231 xenograft growth.* Breast Cancer Res, 2009. **11**(3): p. R32.

7) Reid, D.C., et al., *Next-generation SELEX identifies sequence and structural determinants of splicing factor binding in human pre-mRNA sequence.* RNA, 2009. **15**(12): p. 2385-97.

8) Eom, T., et al., *NOVA-dependent regulation of cryptic NMD exons controls synaptic protein levels after seizure.* Elife, 2013. **2**: p. e00178.

9) Bafico, A., et al., *An autocrine mechanism for constitutive Wnt pathway activation in human cancer cells.* Cancer Cell, 2004. **6**(5): p. 497-506.

10) Cowling, V.H., et al., *c-Myc transforms human mammary epithelial cells through repression of the Wnt inhibitors DKK1 and SFRP1.* Mol Cell Biol, 2007. **27**(14): p. 5135-46.

11) Mikheev, A.M., et al., *Dickkopf-1 mediated tumor suppression in human breast carcinoma cells.* Breast Cancer Res Treat, 2008. **112**(2): p. 263-73.

12) Oskarsson, T., et al., *Breast cancer cells produce tenascin C as a metastatic niche component to colonize the lungs.* Nat Med, 2011. **17**(7): p. 867-74.